# Score matching through the roof: linear, nonlinear, and latent variables causal discovery

## Abstract

Causal discovery from observational data holds great promise, but existing methods rely on strong assumptions about the underlying causal structure, often requiring full observability of all relevant variables. We tackle these challenges by leveraging the score function $\nabla \log p(X)$ of observed variables for causal discovery and propose the following contributions. First, we generalize the existing results of identifiability with the score to additive noise models with minimal requirements on the causal mechanisms. Second, we establish conditions for inferring causal relations from the score even in the presence of hidden variables; this result is two-faced: we demonstrate the score's potential as an alternative to conditional independence tests to infer the equivalence class of causal graphs with hidden variables, and we provide the necessary conditions for identifying direct causes in latent variable models. Building on these insights, we propose a flexible algorithm for causal discovery across linear, nonlinear, and latent variable models, which we empirically validate.

## 1 Introduction

The inference of causal effects from observations holds the potential for great impact arguably in any domain of science, where it is crucial to be able to answer interventional and counterfactual queries from observational data [1, 2, 3]. Existing causal discovery methods can be categorized based on the information they can extract from the data [4], and the assumptions they rely on. Traditional causal discovery methods (e.g. PC, GES [5, 6]) are general in their applicability but limited to the inference of an equivalence class. Additional assumptions on the structural equations generating effects from the cause are, in fact, imposed to ensure the identifiability of a causal order [7, 8, 9, 10]. As a consequence, existing methods for causal discovery require specialized and often untestable assumptions, preventing their application to real-world scenarios.

Further, the majority of existing approaches are hindered by the assumption that all relevant causes of the measured data are observed, which is necessary to interpret associations in the data as causal relationships. Despite the convenience of this hypothesis, it is often not met in practice, and the solutions relaxing this requirement face substantial limitations. The FCI algorithm [11] can only return an equivalence class from the data. Appealing to additional restrictions ensures the identifiability of some direct causal effects in the presence of latent variables: RCD [12] relies on the linear non-Gaussian additive noise model, whereas CAM-UV [13] requires nonlinear additive mechanisms. Nevertheless, the strict conditions on the structural equations hold back their applicability to more general settings.

Our paper tackles these challenges and can be put in the context of a recent line of work that derives a connection between the score function $\nabla \log p(X)$ and the causal graph underlying the data-generating process [14, 15, 16, 17, 18, 19]. The use of the score for causal discovery is practically appealing, as it yields advantages in terms of scalability to high dimensional graphs [16]

Submitted to 38th Conference on Neural Information Processing Systems (NeurIPS 2024). Do not distribute.

and guarantees of finite sample complexity bounds [20]. Instead of imposing assumptions that ensure strong, though often impractical, theoretical guarantees, we organically demonstrate different levels of identifiability based on the strength of the modeling hypotheses, always relying on the score function to encode all the causal information in the data. Starting from results of Spantini et al. [21] and Lin [22], we show how constraints on the Jacobian of the score $\nabla^2 \log p(X)$ can be used as an alternative to conditional independence testing to identify the Markov equivalence class of causal models with hidden variables. Further, we prove that the score function identifies the causal direction of additive noise models, with minimal assumptions on the causal mechanisms. This extends the previous findings of Montagna et al. [17], limited by the assumption of nonlinearity of the causal effects, and Ghoshal and Honorio [14], limited to linear mechanisms. On these results, we build the main contributions of our work, enabling the identification of direct causal effects in hidden variables models.

**Our main contributions** are as follows: *(i)* We present the necessary conditions for the identifiability of direct causal effects and the presence of hidden variables with the score in the case of latent variables models. *(ii)* We propose AdaScore (Adaptive Score-based causal discovery), a flexible algorithm for causal discovery based on score matching estimation of $\nabla \log p(X)$ [23]. Based on the user's belief about the plausibility of several modeling assumptions on the data, AdaScore can output a Markov equivalence class, a directed acyclic graph, or a mixed graph, accounting for the presence of unobserved variables. To the best of our knowledge, the broad class of causal models handled by our method is unmatched by other approaches in the literature.

## 2  Model definition and related works

In this section, we introduce the formalism of structural causal models (SCMs), separately for the the cases with and without hidden variables.

### 2.1  Causal model with observed variables

Let $X$ be a set of random variables in $\mathbb{R}$ defined according to the set of structural equations

$$X_i := f_i(X_{\mathrm{PA}_i^{\mathcal{G}}}, N_i), \quad \forall i = 1, \ldots, k. \tag{1}$$

$N_i \in \mathbb{R}$ are mutually independent random variables with strictly positive density, known as *noise* or *error terms*. The function $f_i$ is the *causal mechanism* mapping the set of *direct causes* $X_{\mathrm{PA}_i^{\mathcal{G}}}$ of $X_i$ and the noise term $N_i$, to $X_i$'s value. A structural causal model (SCM) is defined as the tuple $(X, N, \mathcal{F}, \mathbb{P}_N)$, where $\mathcal{F} = (f_i)_{i=1}^k$ is the set of causal mechanisms, and $\mathbb{P}_N$ is the joint distribution relative to the density $p_N$ over the noise terms $N \in \mathbb{R}^k$. We define the *causal graph* $\mathcal{G}$ as a directed acyclic graph (DAG) with nodes $X = \{X_1, \ldots, X_k\}$, and the set of edges defined as $\{X_j \to X_i : X_j \in X_{\mathrm{PA}_i^{\mathcal{G}}}\}$, such that $\mathrm{PA}_i^{\mathcal{G}}$ are the indices of the parent nodes of $X_i$ in the graph $\mathcal{G}$. (In the remainder of the paper, we adopt the following notation: given a set of random variables $Y = \{Y_1, \ldots, Y_n\}$ and a set of indices $Z \subset \mathbb{N}$, then $Y_Z = \{Y_i | i \in Z, Y_i \in Y\}$.)

Under this model, the probability density of $X$ satisfies the *Markov factorization* (e.g. Peters et al. [1] Proposition 6.31):

$$p(x) = \prod_{i=1}^k p(x_i | x_{\mathrm{PA}_i^{\mathcal{G}}}), \tag{2}$$

where we adopt the convention of lowercase letters referring to realized random variables, and use $p$ to denote the density of different random objects, when the distinction is clear from the argument. This factorization is equivalent to the *global Markov condition* (e.g. Peters et al. [1] Proposition 6.22) that demands that for all $\{X_i, X_j\} \in X, X_Z \subseteq X \setminus \{X_i, X_j\}$, then

$$X_i \perp\!\!\!\perp_{\mathcal{G}}^d X_j | X_Z \implies X_i \perp\!\!\!\perp X_j | X_Z,$$

where $(\cdot \perp\!\!\!\perp \cdot | \cdot)$ denotes probabilistic conditional independence of $X_i, X_j$ given $X_Z$, and $(\cdot \perp\!\!\!\perp_{\mathcal{G}}^d \cdot | \cdot)$ is the notation for *d-separation*, a criterion of conditional independence defined on the graph $\mathcal{G}$ (Definition 5 of the appendix). As it is commonly done, we assume that the reverse direction $X_i \perp\!\!\!\perp X_j | X_Z \implies X_i \perp\!\!\!\perp_{\mathcal{G}}^d X_j | X_Z$ hold, and we say that the density $p$ is *faithful* to the graph $\mathcal{G}$ [2, 24] (hence the *faithfulness assumption*). Together with the global Markov condition, faithfulness implies an equivalence between the probabilistic and graphical notions of conditional independence:

$$X_i \perp\!\!\!\perp X_j | X_Z \iff X_i \perp\!\!\!\perp_{\mathcal{G}}^d X_j | X_Z. \tag{3}$$

In general, several DAGs may entail the same set of d-separations: graphs sharing such common structure form a *Markov equivalence class* (see Definition 6 in the appendix).

The above model assumes that there aren't any unobserved causes of variables in $X$, other than the noise terms in $N$. As we are interested in distributions with potential hidden variables, we will now generalize our model to represent data-generating processes that may involve latent causes.

**Definitions on graphs.**   As graphs play a central role in our work, Appendix A.1 provides a detailed overview of the fundamental notation and definitions that we rely on in the remainder of the paper. For the next section, we advise the reader to be comfortable with the notions of *ancestors* (Definition 2) and *inducing paths* (Definition 3) in DAGs.

**Closely related works.**   Several methods for the causal discovery of fully observable models using the score have been recently proposed. Ghoshal and Honorio [14] demonstrates the identifiability of the linear non-Gaussian model from the score, and it is complemented by Rolland et al. [15], which shows the connection between score matching estimation of $\nabla \log p(X)$ and the inference of causal graphs underlying nonlinear additive noise models with Gaussian noise terms, also allowing for sample complexity bounds [20]. Montagna et al. [17] provides identifiability results in the nonlinear setting, without posing any restriction on the distribution of the noise terms. Montagna et al. [16] is the first to show that the Jacobian of the score provides information equivalent to conditional independence testing in the context of causal discovery, limited to the case of additive noise models. All of these studies make specialized assumptions to find theoretical guarantees of identifiability, whereas our paper provides a unifying view of causal discovery with the score function, which generalizes and expands the existing results.

## 2.2   Causal model with unobserved variables

Under the model (1), we consider the case where the set of variables $X$ is partitioned into the disjoint subsets of *observed* random variables $V = \{V_1, \dots, V_d\}$ and *unobserved* (or *latent*) random variables $U = \{U_1, \dots, U_p\}$. We assume that the following set of structural equations is satisfied:

$$V_i := f_i(V_{\mathrm{PA}_i^{\mathcal{G}}}, U^i, N_i), \quad \forall i = 1, \dots, d, \tag{4}$$

where $U^i$ stands for the set of unobserved parents of $V_i$, and $V_{\mathrm{PA}_i^{\mathcal{G}}} = \{V_k | k \in \mathrm{PA}_i^{\mathcal{G}}, V_k \in V\}$ are the observed direct causes of $V_i$. Some of the causal relations and the conditional independencies implied by the set of equations (4) can be summarized in a graph obtained as a *marginalization* of the DAG $\mathcal{G}$ onto the observable nodes $V$.

**Definition 1** (Marginal graph, Zhang [25])**.** Let $X = V \dot{\cup} U$ and $\mathcal{G}$ be a DAG over $X$. The following construction gives the *marginal* graph $\mathcal{M}_V^{\mathcal{G}}$, with nodes $V$ and edges found as follows:

- pair of nodes $V_i, V_j$ are adjacent in the graph $\mathcal{M}_V^{\mathcal{G}}$ if and only if there is an inducing path between them relative to $U$ in $\mathcal{G}$;

- for each pair of adjacent nodes $V_i, V_j$ in $\mathcal{M}_V^{\mathcal{G}}$, orient the edge as $V_i \rightarrow V_j$ if $V_i$ is an ancestor of $V_j$ in $\mathcal{G}$, else orient it as $V_i \leftrightarrow V_j$.

We define the map $\mathcal{G} \mapsto \mathcal{M}_V^{\mathcal{G}}$ as the *marginalization* of the DAG $\mathcal{G}$ onto $V$, the observable nodes.

The graph resulting from the above construction is a maximal ancestral graph (MAG, Definition 4), hence we will often refer to it as the *marginal MAG* of $\mathcal{G}$. Intuitively, a directed edge denotes the presence of an ancestorship relation, whereas bidirected edges represent dependencies that can not be removed by conditioning on any of the variables in the graph.

In the case of DAGs, d-separation encodes the probabilistic conditional independence relations between the variables of $X$ in the graph $\mathcal{G}$, as explicit by Equation (3). Such notion of graphical separation has a natural generalization to maximal ancestral graphs, known as *m-separation* (Definition 5 of the appendix). Zhang [25] shows that *m-separation* and *d-separation* are in fact equivalent (see Lemma 1 of the appendix), such that given $V_Z \subset V$ and $\{V_i, V_j\} \subset V$, the following holds:

$$V_i \perp\!\!\!\perp_{\mathcal{G}}^{d} V_j | V_Z \setminus \{V_i, V_j\} \iff V_i \perp\!\!\!\perp_{\mathcal{M}_V^{\mathcal{G}}}^{m} V_j | V_Z \setminus \{V_i, V_j\}, \tag{5}$$

where $(\cdot \perp\!\!\!\perp_{\mathcal{M}_V^{\mathcal{G}}}^{m} \cdot | \cdot)$ denotes *m-separation* relative to the graph $\mathcal{M}_V^{\mathcal{G}}$. Just like with DAGs, MAGs that imply the same set of conditional independencies define an equivalence class. Usually, the common structure of these graphs is represented by partial ancestral graphs (PAGs, Definition 7 of the appendix). We use $\mathcal{P}_{\mathcal{M}_V^{\mathcal{G}}}$ to denote the PAG relative to $\mathcal{M}_V^{\mathcal{G}}$.

> **Problem definition.** In this work, our goal is to provide theoretical guarantees for the identifiability of the Markov equivalence class of the marginal graph $\mathcal{M}_V^{\mathcal{G}}$ and its direct causal effects with the score, where variables $V_i$ are defined according to Equation (4).

Without further assumptions on the data-generating process, we can identify the graph $\mathcal{M}_V^{\mathcal{G}}$ only up to its partial ancestral graph, as discussed in the next section.

**Closely related works.** Causal discovery with latent variables have been first studied in the context of *constraint-based* approaches with the FCI algorithm [11], which shows the identifiability of the equivalence class of a marginalized graph via conditional independence testing. The RCD and CAM-UV [12, 13] approaches instead demonstrate the inferrability of directed causal edges via regression and residuals independence testing. Both methods rely on strong assumptions on the causal mechanisms: their theoretical guarantees apply to models where the effects are generated by a linear (RCD) or nonlinear (CAM-UV) additive contribution of each cause. Our work demonstrates that using the score function for causal discovery unifies and generalizes these results, presenting an alternative to conditional independence testing for constraint-based methods, and being agnostic about the class of causal mechanisms of the observed variables, under the weaker requirement of additivity of the noise terms.

## 3 Theory for a score-based test of separation

In this section, we show that for $V \subseteq X$ generated according to Equation (4) the Hessian matrix of $\log p(V)$ identifies the equivalence class of the marginal MAG $\mathcal{M}_V^{\mathcal{G}}$. It has already been proven that cross-partial derivatives of the log-likelihood are informative about a set of conditional independence relationships between random variables: Spantini et al. [21] (Lemma 4.1) shows that, given $V_Z \subseteq X$ such that $\{V_i, V_j\} \subseteq V_Z$, then

$$\frac{\partial^2}{\partial V_i \partial V_j} \log p(V_Z) = 0 \iff V_i \perp\!\!\!\perp V_j | V_Z \setminus \{V_i, V_j\}. \tag{6}$$

Equation (3) resulting from faithfulness and the directed global Markov property immediately implies that this expression can be used as a test of conditional independence to identify the Markov equivalence class of the graph $\mathcal{M}_V^{\mathcal{G}}$, as commonly done in constraint-based causal discovery (for reference, see e.g. Section 3 in Glymour et al. [4]). This result generalizes Lemma 1 of Montagna et al. [16], where it is used to define constraints to infer edges in the causal structure without latent variables.

**Proposition 1** (Adapted[1] from [21]). *Let $V$ be a set of random variables with strictly positive density generated according to model (4). For each set $V_Z \subseteq V$ of nodes in $\mathcal{M}_V^{\mathcal{G}}$ such that $\{V_i, V_j\} \subseteq V_Z$, the following holds for each supported value $v_Z$:*

$$\frac{\partial^2}{\partial V_i \partial V_j} \log p(v_Z) = 0 \iff V_i \perp\!\!\!\perp_{\mathcal{M}_V^{\mathcal{G}}}^{m} V_j | V_Z \setminus \{V_i, V_j\}.$$

The result of Proposition 1 presents an alternative to conditional independence testing in *constraint-based* approaches to causal discovery, showing that the equivalence class of the graph $\mathcal{M}_V^{\mathcal{G}}$ can be identified using the cross partial derivatives of the log-likelihood as a test of conditional independence between variables, much in the spirit of the Fast Causal Inference algorithm [11]. Identifying the

---

[1]In their Lemma 4.1 Spantini et al. [21] provides the connection between vanishing cross-partial derivatives of the log-likelihood and conditional independence of random variables. Note that this result does not depend on the assumption of a generative model, thus holding beyond the set of structural equations (4). Our result adapts their finding to the case when observations are generated according to a fully observable causal model.

156 Markov equivalence class is the most we can hope to achieve without further hypotheses. As we will
157 see in the next section, the score function can also help leverage additional restrictive assumptions on
158 the causal mechanisms of Equation (4) to identify direct causal effects.

## 4 A theory of identifiability from the score

160 In this section, we show that, under additional assumptions on the data-generating process, we can
161 identify the direct causal relations that are not influenced by unobserved variables, as well as the
162 presence of unobserved active paths (Definition 5) between nodes in the marginalized graph $\mathcal{M}_V^{\mathcal{G}}$.

163 As a preliminary step before diving into causal discovery with latent variables, we show how the
164 properties of the score function identify edges in directed acyclic graphs, that is in the absence of
165 latent variables (when $U = \emptyset$ and $\mathcal{G} = \mathcal{M}_V^{\mathcal{G}}$). The goal of the next section is two-sided: first, it
166 introduces the fundamental ideas connecting the score function to causal discovery that also apply to
167 hidden variable models, second, it extends the existing theory of causal discovery with score matching
168 to additive noise models with both linear and nonlinear mechanisms.

### 4.1 Warm up: identifiability without latent confounders

170 In this section, we summarise and extend the theoretical findings presented in Montagna et al. [17],
171 where the authors show how to derive constraints on the score function that identify the causal order of
172 the DAG $\mathcal{G}$ where all the variables in the set $X$ are observed. Define the structural relations of (1) as:

$$X_i \coloneqq h_i(X_{\mathrm{PA}_i^{\mathcal{G}}}) + N_i, i = 1, \ldots, k, \tag{7}$$

173 with three times continuously differentiable mechanisms $h_i$, noise terms centered at zero, and strictly
174 positive density $p_X$. Given the Markov factorization of Equation (2), the components of the score
175 function $\nabla \log p(x)$ are:

$$
\begin{aligned}
\partial_{X_i} \log p(x) &= \partial_{X_i} \log p(x_i | x_{\mathrm{PA}_i^{\mathcal{G}}}) + \sum_{j \in \mathrm{CH}_i^{\mathcal{G}}} \partial_{X_i} \log p(x_j | x_{\mathrm{PA}_j^{\mathcal{G}}}) \\
&= \partial_{N_i} \log p(n_i) - \sum_{j \in \mathrm{CH}_i^{\mathcal{G}}} \partial_{X_i} h_j(x_{\mathrm{PA}_j^{\mathcal{G}}}) \partial_{N_j} \log p(n_j),
\end{aligned}
\tag{8}
$$

176 where $\mathrm{CH}_i^{\mathcal{G}}$ denotes the set of children of node $X_i$. We observe that if a node $X_s$ is a *sink*, i.e. a
177 node satisfying $\mathrm{CH}_s^{\mathcal{G}} = \emptyset$, then the summation over the children vanishes, implying that:

$$\partial_{X_s} \log p(x) = \partial_{N_s} \log p(n_s). \tag{9}$$

178 The key point is that the score component of a sink node is a function of its structural equation noise
179 term, such that one could learn a consistent estimator of $\partial_{X_s} \log p_X$ from a set of observations of the
180 noise term $N_s$. Given that, in general, one has access to $X$ samples rather than observations of the
181 noise random variables, authors in Montagna et al. [17] show that $N_s$ of a sink node can be consistently
182 estimated from i.i.d. realizations of $X$. For each node $X_1, \ldots, X_k$, we define the quantity:

$$R_i \coloneqq X_i - \mathbf{E}[X_i | X_{\setminus X_i}], \tag{10}$$

183 where $X_{\setminus X_i}$ are the random variables in the set $X \setminus \{X_i\}$. $\mathbf{E}[X_i | X_{\setminus X_i}]$ is the optimal least squares
184 predictor of $X_i$ from all the remaining nodes in the graph, and $R_i$ is the regression residual. For
185 a sink node $X_s$, the residual satisfies:

$$R_s = N_s, \tag{11}$$

186 which can be seen by rewriting $\mathbf{E}[X_s | X_{\setminus X_s}] = h_s(X_{\mathrm{PA}^{\mathcal{G}}}) + \mathbf{E}[N_s | X_{\mathrm{DE}_s^{\mathcal{G}}}, X_{\mathrm{ND}_s^{\mathcal{G}}}] =$
187 $h_s(X_{\mathrm{PA}^{\mathcal{G}}}) + \mathbf{E}[N_s]$, where $X_{\mathrm{DE}_s^{\mathcal{G}}}$ and $X_{\mathrm{ND}_s^{\mathcal{G}}}$ denotes the descendants and non-descendants of $X_s$,
188 respectively. Equations (9) and (11) together imply that the score $\partial_{N_s} \log p(N_s)$ is a function of $R_s$,
189 such that it is possible to find a consistent approximator of the score of a sink from observations of $R_s$.

190 **Proposition 2** (Generalization of Lemma 1 in Montagna et al. [17])**.** *Let $X$ be a set of random*
191 *variables, generated by a restricted additive noise model (Definition 9) with structural equations* (7)*,*
192 *and let $X_j \in X$. Consider $r_j$ in the support of $R_j$. Then:*

$$X_j \text{ is a sink} \iff \mathbf{E}\left[\left(\mathbf{E}\left[\partial_{X_j} \log p(X) \mid R_j = r_j\right] - \partial_{X_j} \log p(X)\right)^2\right] = 0. \tag{12}$$

Our result generalizes Lemma 1 in Montagna et al. [17], as they assume $X$ generated by an identifiable additive noise model with nonlinear mechanisms. Instead, we remove the nonlinearity assumption and make the weaker hypothesis of a *restricted* additive noise model, which is provably identifiable [9], in the formal sense defined in the appendix (Definition 8). This result doesn't come as a surprise, given the previous findings of Ghoshal and Honorio [14] showing that the score infers linear non-Gaussian additive noise models: Proposition 2 provides a unifying and general theory for the identifiability of models with potentially mixed linear and nonlinear mechanisms.

Based on these insights, Montagna et al. [17] propose the NoGAM algorithm to exploit the condition in (12) for identifying the causal order of the graph: being $\mathbf{E}\left[\partial_{X_i} \log p(X) \mid R_i\right]$ the optimal least squares estimator of the score of node $X_i$ from $R_i$, a sink node is characterized as the $\operatorname{argmin}_i \mathbf{E}\left[\mathbf{E}\left[\partial_{X_i} \log p(X) \mid R_i\right] - \partial_{X_i} \log p(X)\right]^2$, where in practice the residuals $R_i$, the score components and the least squares estimators are replaced by their empirical counterparts. After a sink node is identified, it is removed from the graph and assigned a position in the order, and the procedure is iteratively repeated up to the source nodes. Being the score estimated by score matching techniques [23], we usually make reference to *score matching-based* causal discovery.

In the next section, we show how we can generalize these results to identify direct causal effects between a pair of variables in the marginal MAG $\mathcal{M}_V^{\mathcal{G}}$ when $U \neq \emptyset$

## 4.2 Identifiability in the presence of latent confounders

We now introduce the last of our main theoretical results, that is: given a pair of nodes $V_i$, $V_j$ that are adjacent in the graph $\mathcal{M}_V^{\mathcal{G}}$ with $U \neq \emptyset$, we can use the score function to identify the presence of a direct causal effect between $V_i$ and $V_j$, or that of an active path that is influenced by unobserved variables. Given that the causal model of Equation (4) ensures identifiability only up to the equivalence class, we need additional restrictive assumptions. In particular, we enforce an additive noise model with respect to both the observed and unobserved noise variables. This corresponds to an additive noise model on the observed variables with the noise terms recentered by the latent causal effects.

**Assumption 1** (SCM assumptions). *The set of structural equations of the observable variables specified in* (4) *is now defined as:*

$$V_i \coloneqq f_i(V_{\mathrm{PA}_i^{\mathcal{G}}}) + g_i(U^i) + N_i, \forall i = 1, \ldots, d, \tag{13}$$

*assuming the mechanisms $f_i$ to be of class $\mathcal{C}^3(\mathbb{R}^{|V_{\mathrm{PA}_i^{\mathcal{G}}}|})$, and mutually independent noise terms with strictly positive density function. The $N_i$'s are assumed to be non-Gaussian when $f_i$ is linear in some of its arguments.*

Crucially, our hypothesis is weaker than those required by two state-of-the-art approaches, CAM-UV [13] and RCD [12]: CAM-UV assumes a Causal Additive Model (CAM) with structural equations with nonlinear mechanisms in the form $V_i \coloneqq \sum_{k \in \mathrm{PA}_i^{\mathcal{G}}} f_{ik}(V_k) + \sum_{U_k^i} g_{ik}(U_k^i) + N_i$, and RCD requires an additive noise model with linear effects of both the latent and observed causes. Thus, our model encompasses and extends the nonlinear and linear settings of CAM-UV and RCD, such that the theory developed in the remainder of the section is valid for a broader class of causal models.

Our first step is rewriting the structural relations in (13) as:

$$\begin{aligned} V_i &\coloneqq f_i(V_{\mathrm{PA}_i^{\mathcal{G}}}) + \tilde{N}_i, \\ \tilde{N}_i &\coloneqq g_i(U^i) + N_i, \forall i = 1, \ldots, d, \end{aligned} \tag{14}$$

which provides an additive noise model in the form of (7). Next, we define the following regression residuals for any node $V_k$ in the graph $\mathcal{M}_V^{\mathcal{G}}$:

$$R_k(V_Z) \coloneqq V_k - \mathbf{E}[V_k \mid V_{Z \setminus \{k\}}], \tag{15}$$

where $V_{Z \setminus \{k\}}$ denotes the set of random variables $V_Z \setminus \{V_k\}$.

Given these definitions, we are ready to show how directed edges, and the presence of unobserved variables can be identified from the score of linear and nonlinear additive noise models.

### 4.2.1 Identifiability of directed edges

Consider $V_i, V_j$ adjacent nodes in the PAG $\mathcal{P}_{\mathcal{M}_V^{\mathcal{G}}}$: we want to investigate when a direct causal effect $V_i \in V_{\mathrm{PA}_j^{\mathcal{G}}}$ can be identified from the score. We make the following observations: for $V_Z = V_{\mathrm{PA}_j^{\mathcal{G}}} \cup \{V_j\}$ and $V_{\mathrm{PA}_j^{\mathcal{G}}} \perp\!\!\!\perp_d^{\mathcal{G}} U^j$, by Equation (15) it follows

$$R_j(V_Z) = \tilde{N}_j - \mathbf{E}[\tilde{N}_j], \tag{16}$$

where we use $V_{\mathrm{PA}_j^{\mathcal{G}}} \perp\!\!\!\perp_d^{\mathcal{G}} U^j$ to write $\mathbf{E}[\tilde{N}_j | V_{Z \setminus \{j\}}] = \mathbf{E}[\tilde{N}_j]$. Moreover, we note that $V_j$ is a sink node relative to $\mathcal{M}_{V_Z}^{\mathcal{G}}$, the marginalization of $\mathcal{G}$ onto $V_Z$. In analogy to the case without latent variables, we can show that $\partial_{V_j} \log p(V_Z)$ is a function of $\tilde{N}_j$, the error term in the additive noise model of Equation (14), such that the score of $V_j$ can be consistently predicted from observations of the residual $R_j(V_Z)$.

**Proposition 3.** *Let X be generated by a restricted additive noise model with structural equations* (7), *and causal graph $\mathcal{G}$. Consider $V_i, V_j$ adjacent in $\mathcal{M}_V^{\mathcal{G}}$, marginalization of $\mathcal{G}$. Further, assume that the score component $\partial_{V_j} \log p(V_Z)$ is not constant for uncountable values of $V_Z$.*

*(i) Let $V_Z = V_{\mathrm{PA}_j^{\mathcal{G}}} \cup \{V_i, V_j\}$, and $r_j \in \mathbb{R}$ in the support of $R_j(V_Z)$. Then:*

$$V_{\mathrm{PA}_j^{\mathcal{G}}} \perp\!\!\!\perp_{\mathcal{G}}^d U^j \wedge V_i \in V_{\mathrm{PA}_j^{\mathcal{G}}} \iff \mathbf{E}[\partial_{V_j} \log p(V_Z) - \mathbf{E}[\partial_{V_j} \log p(V_Z) | R_j(V_Z) = r_j]]^2 = 0.$$

*(ii) Let $V_Z \subseteq V$, such that $\{V_i, V_j\} \subseteq V_Z$. Then:*

$$V_{\mathrm{PA}_j^{\mathcal{G}}} \not\perp\!\!\!\perp_{\mathcal{G}}^d U^j \vee V_i \notin V_{\mathrm{PA}_j^{\mathcal{G}}} \iff \mathbf{E}[\partial_{V_j} \log p(V_Z) - \mathbf{E}[\partial_{V_j} \log p(V_Z) | R_j(V_Z) = r_j]]^2 \neq 0.$$

Intuitively, the proposition has two essential implications. Part *(i)* provides the condition for the identifiability of the potential direct causal effect between a pair $V_i, V_j$, that is, when the association between $V_j$ and its observed parents is not influenced by active paths that involve latent variables. This condition is necessary: given an active path such that $V_{\mathrm{PA}_j^{\mathcal{G}}} \not\perp\!\!\!\perp_{\mathcal{G}}^d U^j$, the score could not identify a direct causal effect $V_i \to V_j$, which is the content of the second part of the proposition.

We have established theoretical guarantees of identifiability for linear and nonlinear additive noise models, even in the presence of hidden variables: we find that the score function is a means for the identifiability of all direct parental relations that are not influenced by unobserved variables; all the remaining arrowheads of the edges in the graph $\mathcal{M}_V^{\mathcal{G}}$ are identified no better than in the equivalence class. Based on these insights, we propose AdaScore, a score matching-based algorithm for the inference of Markov equivalence classes, direct causal effects, and the presence of latent variables.

### 4.3 A score-based algorithm for causal discovery

Building on our theory, we propose AdaScore, a generalization of NoGAM to linear and nonlinear additive noise models with latent variables. The main strength of our approach is its adaptivity with respect to structural assumptions: based on the user's belief about the plausibility of several modeling assumptions on the data, AdaScore can output an equivalence class (using the condition of Proposition 1 instead of conditional independence testing in an FCI-like algorithm), a directed acyclic graph (as in NoGAM), or a mixed graph, accounting for the presence of unobserved variables. We now describe the version of our algorithm whose output is a mixed graph, where we rely on score matching estimation of the score and its Jacobian (Appendix C.2). At an intuitive level, we find unoriented edges using Proposition 1, i.e. checking for dependencies in the form of non-zero entries in the Jacobian of the score via hypothesis testing on the mean, and find the edges' directions via the condition of Proposition 3, i.e. by estimating residuals of each node $X_i$ and checking whether they can correctly predict the $i$-th score entry (the vanishing mean squared errors are verified by hypothesis test of zero mean). It would be tempting to simply find the skeleton (i.e. the graphical representation of the constraints of an equivalence class) first via the well-known adjacency search of the FCI algorithm and then iterate through all neighborhoods of all nodes to orient edges using Proposition 3. This would be prohibitively expensive, as finding the skeleton is well-known to have super-exponential computational complexity [11]. Instead, we propose an alternative solution: exploiting the fact that some nodes may not be influenced by latent variables, we first use Proposition 2 to find sink nodes

that are not affected by latents (using hypothesis testing to find vanishing mean squared error in the score predictions from the residuals), in the spirit of the NoGAM algorithm. If there is such a sink, we search all its adjacent nodes via Proposition 1 (plus an optional pruning step for better accuracy, Appendix C.2), and orient the inferred edges towards the sink. Else, if no sink can be found, we pick a node in the graph and find its neighbors by Proposition 1, orienting its edges using the condition in Proposition 3 (score estimation by residuals under latent effects). This way, we get an algorithm that is polynomial in the best case (Appendix C.3). Details on AdaScore are provided in Appendix C, while a pseudo-code summary is provided in the Algorithm 1 box.

---

**Algorithm 1** Simplified pseudo-code of AdaScore

---

   **while** nodes remain **do**
      **if** Proposition 3 finds a sink with all parents observed **then**
         add edges from adjacent nodes to sink
      **else**
         pick some remaining node $V_i \in V$
         prune neighbourhood of $V_i$ using Proposition 1
         orient edges adjacent to $V_i$ using Proposition 3
         **if** $V_i$ has outgoing directed edge to some $V_j \in V$ **then**
            **continue with** $V_j$
         **else**
            remove $V_i$ form remaining nodes
   prune remaining bidirected edges using Proposition 1

---

# 5 Experiments

We use the `causally`[2] Python library [26] to generate synthetic data with known ground truths, created as Erdös-Rényi sparse and dense graphs, respectively with probability of edge between pair of nodes equals $0.3$ and $0.5$. We sample the data according to linear and nonlinear mechanisms with additive noise, where the nonlinear functions are parametrized by a neural network with random weights, a common approach in the literature [18, 26, 27, 28, 29]. Noise terms are sampled from a uniform distribution in the $[-2, 2]$ range. Hidden causal effects are obtained by randomly picking two nodes and dropping the corresponding column from the data matrix. See Appendix D.1 for further details on the data generation. As metric, we consider the structural Hamming distance (SHD) [30, 31], a simple count of the number of incorrect edges, where missing and wrongly directed edges count as one error. We fix the level of the hypothesis tests of AdaScore to $0.05$, which is a common choice in the absence of prior knowledge. We compare AdaScore to NoGAM, CAM-UV, RCD, and DirectLiNGAM, whose assumptions are detailed in Table 1. In the main manuscript, we comment on the results on datasets of $1000$ observations from *dense* graphs, with and without latent variables. Additional experiments including those on sparse networks are presented in Appendix E. Our synthetic data are standardized by their empirical variance to remove shortcuts in the data [18, 32].

**Discussion.** Our experimental results on models without latent variables of Figure 1a show that when causal relations are linear, AdaScore can recover the causal graph with accuracy that is comparable with all the other benchmarks, with the exception of DirectLiNGAM. On nonlinear data AdaScore presents better performance than CAM-UV, RCD, and DirectLiNGAM while being comparable to NoGAM in accuracy. This is in line with our expectations: in the absence of finite sample errors and in the fully observable setting, NoGAM and AdaScore are indeed the same algorithms. When inferring under latent causal effects, Figure 1b, our method performs comparably to CAM-UV and RCD on graphs up to seven nodes while slightly degrading on nine nodes. Additionally, AdaScore outperforms NoGAM in this setting, as we would expect according to our theory. Overall, we observe that our method is robust to a variety of structural assumptions, with accuracy that is often comparable and sometimes better than competitors (as in nonlinear observable settings). We remark that although AdaScore does not clearly outperform the other baselines, its broad theoretical guarantees of identifiability are not matched by any available method in the literature; this makes it an appealing option for inference in realistic scenarios that are hard to investigate with synthetic data, where the structural assumptions of the causal model underlying the observations are unknown.

---

[2]`https://causally.readthedocs.io/en/latest/`

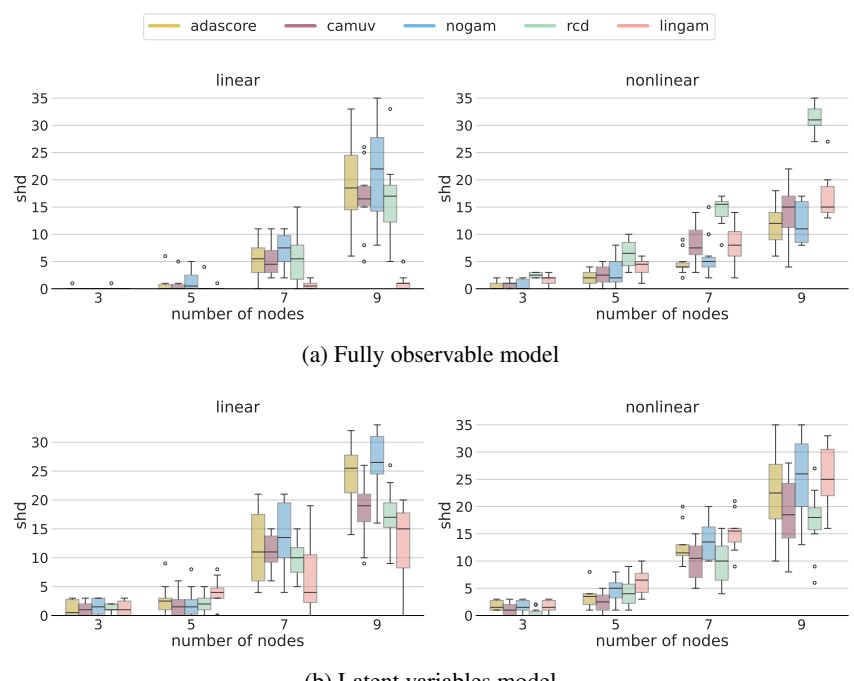

(a) Fully observable model

(b) Latent variables model

Figure 1: Empirical results on dense graphs with different numbers of nodes, on fully observable (no hidden variables) and latent variable models. We report the SHD accuracy (the lower, the better). We note that DirectLiNGAM is surprisingly robust to different structural assumptions, and AdaScore is generally comparable or better (as in nonlinear observable data) than the other benchmarks.

Table 1: Experiments causal discovery algorithms. The content of the cells denotes whether the method supports (✓) or not (✗) the condition specified in the corresponding row.

|  | CAM-UV | RCD | NoGAM | DirectLiNGAM | AdaScore |
|---|---|---|---|---|---|
| Linear additive noise model | ✗ | ✓ | ✗ | ✓ | ✓ |
| Nonlinear additive noise model | ✗ | ✗ | ✓ | ✗ | ✓ |
| Nonlinear CAM | ✓ | ✗ | ✓ | ✗ | ✓ |
| Latent variables effects | ✓ | ✓ | ✗ | ✗ | ✓ |
| Output | Mixed | Mixed | DAG | DAG | Mixed |

## 6 Conclusion

The existing literature on causal discovery shows a connection between score matching and structure learning in the context of nonlinear ANMs: in this paper, (i) we formalize and extend these results to linear SCMs, and (ii) we show that the score retains information on the causal structure even in the presence of unobserved variables. Additionally, while previous works posit the accent on finding the causal order through the score, we study its potential to identify the Markov equivalence class with a *constraint-based* strategy that does not explicitly require tests of conditional independence, as well as to identify direct causal effects. Our theoretical insights result in AdaScore: unlike existing approaches for the estimation of causal directions, our algorithm provides theoretical guarantees for a broad class of identifiable models, namely linear and nonlinear, with additive noise, in the presence of latent variables. Even though AdaScore does not clearly outperform the existing baselines on our synthetic benchmark, its adaptivity to different structural hypotheses is a step towards causal discovery that is less reliant on prior assumptions, which are often untestable and thus hindering reliable inference in real-world problems. While we do not touch on the task of causal representation learning [33], where causal variables are learned from data, we believe this is a promising research direction in relation to our work due to the specific interplay between score-matching estimation and generative models.

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

# A Useful results

In this section, we provide a collection of results and definitions relevant to the theory of this paper.

## A.1 Definitions over graphs

Let $X = X_1, \ldots, X_d$ a set of random variables. A graph $\mathcal{G} = (X, E)$ consists of finitely many nodes or vertices $X$ and edges $E$. We now provide additional definitions, separately for directed acyclic and mixed graphs.

**Directed acyclic graph.** In a *directed graph*, nodes can be connected by a *directed edge* ($\rightarrow$), and between each pair of nodes there is at most one directed edge. We say that $X_1$ is a *parent* of $X_j$ if $X_i \rightarrow X_j \in E$, in which case we also say that $X_j$ is a *child* of $X_i$. Two nodes are *adjacent* if they are connected by an edge. Three nodes are called a *v-structure* if one node is a child of the other two, e.g. as $X_i \rightarrow X_k \leftarrow X_j$ is a collider. A *path* in $\mathcal{G}$ is a sequence of at least two distinct vertices $X_{i_1}, \ldots, X_{i_m}$ such that there is an edge between $X_{i_k}$ and $X_{i_{k+1}}$. If $X_{i_k} \rightarrow X_{i_{k+1}}$ for every node in the path, we speak of a *directed path*, and call $X_{i_k}$ an *ancestor* of $X_{i_{k+1}}$, $X_{i_{k+1}}$ a *descendant* of $X_{i_k}$. Given the set $\mathrm{DE}_i^{\mathcal{G}}$ of descendants of a node $X_i$, we define the set of *non-descendants* of $X_i$ as $\mathrm{ND}_i^{\mathcal{G}} = X \setminus (\mathrm{DE}_i^{\mathcal{G}} \cup \{X_i\})$. A node without parents is called a *source node*. A node without children is called a *sink node*. A *directed acyclic graph* is a directed graph with no cycles.

**Mixed graph.** In a *mixed graph* nodes can be connected by a *directed edge* ($\rightarrow$) or a *bidirected edge* ($\leftrightarrow$), and between each pair of nodes there is at most one directed edge. Two vertices are said to be *adjacent* in a graph if there is an edge (of any kind) between them. The definitions of *parent*, *child*, *ancestor*, *descendant*, *path* provided for directed acyclic graph also apply in the case of mixed graphs. Additionally, $X_i$ is a spouse of $X_j$ (and vice-versa) if $X_i \leftrightarrow X_j \in E$. An *almost directed cycle* occurs when $X_i \leftrightarrow X_j \in E$ and $X_i$ is an ancestor of $X_j$ in $\mathcal{G}$.

For ease of reference from the main text, we separately provide the definition of inducing paths and ancestors in directed acyclic graphs.

**Definition 2** (Ancestor). Consider a DAG $\mathcal{G}$ with set of nodes $X$, and $X_i, X_j$ elements of $X$. We say that $X_i$ is an *ancestor* of $X_j$ if there is a directed path from $X_i$ to $X_j$ in the graph, as in $X_i \rightarrow \ldots \rightarrow X_j$.

**Definition 3** (Inducing path). Consider a DAG $\mathcal{G}$ with set of nodes $X$, and $Y, Z$ disjoint subsets such that $X = Y \dot{\cup} Z$. We say that there is an *inducing path relative to $Z$* between the nodes $Y_i, Y_j$ if every node on the path that is not in $Z \cup \{Y_i, Y_j\}$ is a collider on the path (i.e. for each $Y_k \in Y$ on the path the sequence $Y_i \ldots \rightarrow Y_k \leftarrow \ldots Y_j$ appears) and every collider on the path is an ancestor of $Y_i$ or $Y_j$.

One natural way to encode inducing paths and ancestral relationships between variables is represented by maximal ancestral graphs.

**Definition 4** (MAG). A *maximal ancestral graph* (MAG) is a mixed graph such that:

1. there are no directed cycles and no almost directed cycles;

2. there are no inducing paths between two non-adjacent nodes.

Next, we define conditional independence in the context of graphs.

**Definition 5** (m-separation). Let $\mathcal{M}$ be a mixed graph with nodes $X$. A path $\pi$ in $\mathcal{M}$ between $X_i, X_j$ elements of $X$ is *active* w.r.t. $Z \subseteq X \setminus \{X_i, X_j\}$ if:

1. every non-collider on $\pi$ is not in $Z$

2. every collider on $\pi$ is an ancestors of a node in $Z$.

$X_i$ and $X_j$ are said to be *m-separated* by $Z$ if there is no active path between $X_i$ and $X_j$ relative to $Z$. Two disjoint sets of variables $W$ and $Y$ are *m-separated* by $Z$ if every variable in $W$ is m-separated from every variable in $Y$ by $Z$.

If m-separation is applied to DAGs, it is called *d-separation*.

The set of directed acyclic graphs that satisfy the same set of conditional independencies form an equivalence class, known as the *Markov equivalence class*.

**Definition 6** (Markov equivalence class of a DAG). Let $\mathcal{G}$ be a DAG with nodes $X$. We denote with $[\mathcal{G}]$ the *Markov equivalence class* of $\mathcal{G}$. A DAG $\tilde{\mathcal{G}}$ with nodes $X$ is in $[\mathcal{G}]$ if the following conditions are satisfied for each pair $X_i, X_j$ of distinct nodes in $X$:

- there is an edge between $X_i, X_j$ in $\mathcal{G}$ if and only if there is an edge between $X_i, X_j$ in $\tilde{\mathcal{G}}$;

- let $Z \subseteq X \setminus \{X_i, X_j\}$. Then $X_i \perp\!\!\!\perp_{\mathcal{G}}^{d} X_j | Z \iff X_i \perp\!\!\!\perp_{\tilde{\mathcal{G}}}^{d} X_j | Z$;

- let $\pi$ be a path between $X_i$ and $X_j$. $X_k$ is a collider in the path $\pi$ in $\mathcal{G}$ if and only if it is a collider in the path $\pi$ in $\tilde{\mathcal{G}}$.

In summary, graphs in the same equivalence class share the edges up to direction, the set of d-separations, and the set of colliders.

Just as for DAGs, there may be several MAGs that imply the same conditional independence statements. Denote the *Markov-equivalence class* of a MAG $\mathcal{M}$ with $[\mathcal{M}]$: this is represented by a partial mixed graph, the class of graphs that can contain four kinds of edges: $\rightarrow$, $\leftrightarrow$, $\circ\!\!-\!\!\circ$ and $\circ\!\!\rightarrow$, and hence three kinds of end marks for edges: arrowhead ($>$), tail ($-$) and circle ($\circ$).

**Definition 7** (PAG, Definition 3 of Zhang [25]). Let $[\mathcal{M}]$ be the Markov equivalence class of an arbitrary MAG $\mathcal{M}$. The partial ancestral graph (PAG) for $[\mathcal{M}]$, $P_{\mathcal{M}}$, is a partial mixed graph such that:

- $P_{\mathcal{M}}$ has the same adjacencies as $\mathcal{M}$ (and any member of $[\mathcal{M}]$) does;

- A mark of arrowhead is in $P_{\mathcal{M}}$ if and only if it is shared by all MAGs in $[\mathcal{M}]$; and

- A mark of tail is in $P_{\mathcal{M}}$ if and only if it is shared by all MAGs in $[\mathcal{M}]$.

Intuitively, a PAG represents an equivalence class of MAGs by displaying all common edge marks shared by all members of the class and displaying circles for those marks that are not in common.

## A.2 Equivalence between m-separation and d-separation

In this section, we provide a proof for equation (5), stating the equivalence between m-separation and d-separation in a formal sense.

**Lemma 1** (Adapted from Zhang [25]). *Let $\mathcal{G}$ be a DAG with nodes $X = V \cup U$, with $V$ and $U$ disjoint sets, and $\mathcal{M}_V^{\mathcal{G}}$ the marginalization of $\mathcal{G}$ onto $V$. For any $\{V_i, V_j\} \in V$ and $V_Z \subseteq V \setminus \{V_i, V_j\}$, the following equivalence holds:*

$$V_i \perp\!\!\!\perp_{\mathcal{G}}^{d} V_j | V_Z \iff V_i \perp\!\!\!\perp_{\mathcal{M}_V^{\mathcal{G}}}^{m} V_j | V_Z.$$

*Proof.* The implication $V_i \perp\!\!\!\perp_{\mathcal{G}}^{d} V_j | V_Z \implies V_i \perp\!\!\!\perp_{\mathcal{M}_V^{\mathcal{G}}}^{m} V_j | V_Z$ is a direct consequence of Lemma 18 from Spirtes and Richardson [34], where we set $S = \emptyset$, since we do not consider selection bias. The implication $V_i \perp\!\!\!\perp_{\mathcal{G}}^{d} V_j | V_Z \impliedby V_i \perp\!\!\!\perp_{\mathcal{M}_V^{\mathcal{G}}}^{m} V_j | V_Z$ follows from Lemma 17 by Spirtes and Richardson [34], again with $S = \emptyset$. Note, that in their terminology "d-separation in MAGs" is what we call m-separation. $\qquad\square$

## A.3 Additive noise model identifiability

We study the identifiability of the additive noise model, reporting results from Peters et al. [9]. We start with a formal definition of identifiability in the context of causal discovery.

**Definition 8** (Identifiable causal model). Let $(X, N, \mathcal{F}, p_N)$ be an SCM with underlying graph $\mathcal{G}$ and $p_X$ joint density function of the variables of $X$. We say that the model is *identifiable* from observational data if the distribution $p_X$ can not be generated by a structural causal model with graph $\tilde{\mathcal{G}} \neq \mathcal{G}$.

First, we consider the case of models of two random variables

$$X_2 := f(X_1) + N, \quad X_1 \perp\!\!\!\perp N. \tag{17}$$

**Condition 1** (Condition 19 of Peters et al. [9]). Consider an additive noise model with structural equations (17). The triple $(f, p_{X_1}, p_N)$ does not solve the following differential equation for all pairs $x_1, x_2$ with $f'(x_2)\nu''(x_2 - f(x_1)) \neq 0$:

$$\xi''' = \xi'' \left( \frac{f''}{f'} - \frac{\nu''' f'}{\nu''} \right) + \frac{\nu''' \nu' f'' f'}{\nu''} - \frac{\nu'(f'')^2}{f'} - 2\nu'' f'' f' + \nu' f''', \tag{18}$$

Here, $\xi := \log p_{X_1}$, $\nu := \log p_N$, the logarithms of the strictly positive densities. The arguments $x_2 - f(x_1)$, $x_1$, and $x_1$ of $\nu$, $\xi$ and $f$ respectively, have been removed to improve readability.

Next, we show that a structural causal model satisfying Condition 1 is identifiable, as in Definition 8

**Theorem 1** (Theorem 20 of Peters et al. [9]). *Let $p_{X_1, X_2}$ the joint distribution of a pair of random variables generated according to the model of equation* (17) *that satisfies Condition 1, with graph $\mathcal{G}$. Then, $\mathcal{G}$ is identifiable from the joint distribution.*

Finally, we show an important fact, holding for identifiable bivariate models, which is that the score $\frac{\partial}{\partial X_1} \log p(x_1, x_2)$ is nonlinear in $x_1$.

**Lemma 2** (Sufficient variability of the score). *Let $p_{X_1, X_2}$ the joint distribution of a pair of random variables generated according to a structural causal model that satisfies Condition 1, with graph $\mathcal{G}$. Then:*

$$\frac{\partial}{\partial X_1}(\xi'(x_1) - f'(x_1)\nu'(x_2 - f(x_1))) \neq 0,$$

*for all pairs $(x_1, x_2)$.*

*Proof.* By contradiction, assume that there exists $(x_1, x_2)$ such that $\frac{\partial}{\partial X_1}(\xi'(x_1) - f'(x_1)\nu'(x_2 - f(x_1))) = 0$. Then:

$$\frac{\partial}{\partial X_1} \left( \frac{\frac{\partial^2}{\partial X_1^2} \pi(x_1, x_2)}{\frac{\partial^2}{\partial X_1 \partial X_2} \pi(x_1, x_2)} \right) = 0,$$

where $\pi(x_1, x_2) = \log p(x_1, x_2)$. By explicitly computing all the partial derivatives of the above equation, we obtain that equation 18 is satisfied, which violates Condition 1. $\square$

These results guaranteeing the identifiability of the bivariate additive noise model can be generalized to the multivariable case, with a set of random variables $X = \{X_1, \ldots, X_k\}$ that satisfy:

$$X_i := f_i(X_{\mathrm{PA}_i^{\mathcal{G}}}) + N_i, i = 1, \ldots, k, \tag{19}$$

where $\mathcal{G}$ is the resulting causal graph directed and acyclic. The intuition is that, rather than studying the multivariate model as a whole, we need to ensure that Condition 1 is satisfied for each pair of nodes, adding restrictions on their marginal conditional distribution.

**Definition 9** (Definition 27 of Peters et al. [9]). Consider an additive noise model with structural equations (19). We call this SCM a *restricted additive noise model* if for all $X_j \in X$, $X_i \in X_{\mathrm{PA}_j^{\mathcal{G}}}$, and all sets $X_S \subseteq X$, $S \subset \mathbb{N}$, with $X_{\mathrm{PA}_j^{\mathcal{G}}} \setminus \{X_i\} \subseteq X_S \subseteq X_{\mathrm{ND}_j}^{\mathcal{G}} \setminus \{X_i, X_j\}$, there is a value $x_S$ with $p(x_S) > 0$, such that the triplet

$$(f_j(x_{\mathrm{PA}_j^{\mathcal{G}} \setminus \{i\}}, \cdot), p_{X_i | X_S = x_S}, p_{N_j})$$

satisfies Condition 1. Here, $f_j(x_{\mathrm{PA}_j^{\mathcal{G}} \setminus \{i\}}, \cdot)$ denotes the mechanism function $x_i \mapsto f_j(x_{\mathrm{PA}_j^{\mathcal{G}}})$. Additionally, we require the noise variables to have positive densities and the functions $f_j$ to be continuous and three times continuously differentiable.

Then, for a restricted additive noise model, we can identify the graph from the distribution.

**Theorem 2** (Theorem 28 of Peters et al. [9]). *Let $X$ be generated by a restricted additive noise model with graph $\mathcal{G}$, and assume that the causal mechanisms $f_j$ are not constant in any of the input arguments, i.e. for $X_i \in X_{\mathrm{PA}_j^{\mathcal{G}}}$, there exist $x_i \neq x_i'$ such that $f_j(x_{\mathrm{PA}_j^{\mathcal{G}} \setminus \{i\}}, x_i) \neq f_j(x_{\mathrm{PA}_j^{\mathcal{G}} \setminus \{i\}}, x_i')$. Then, $\mathcal{G}$ is identifiable.*

## A.4 Other auxiliary results

We state several results that hold for a pair of random variables that are not connected by an active path that includes unobserved variables (active paths are introduced in Definition 5). For the remainder of the section, let $V, U$ be a pair of disjoint sets of random variables, $X = V \cup U$ generated according to the structural causal model defined by the set of equations (1), $\mathcal{G}$ the associated causal graph, and $\mathcal{M}_V^{\mathcal{G}}$ the marginalization onto $V$.

The first statement provides under which condition the unobserved parents of two variables in the marginal MAG are mutually independent random vectors.

**Lemma 3.** *Let $V_j \in V$, and $Z \subset \mathbb{N}$ such that $V_Z = V_{\mathrm{PA}_j^{\mathcal{G}}} \cup \{V_j\}$. Assume $V_{\mathrm{PA}_j^{\mathcal{G}}} \perp\!\!\!\perp_{\mathcal{G}}^d U^j$. Then $U^j \perp\!\!\!\perp_{\mathcal{G}}^d U^{Z_k}$ for each index $Z_k \neq j$.*

*Proof.* The assumption $V_{\mathrm{PA}_j^{\mathcal{G}}} \perp\!\!\!\perp_{\mathcal{G}}^d U^j$ implies that there is no active path in $\mathcal{G}$ between nodes in $V_{\mathrm{PA}_j^{\mathcal{G}}}$ and nodes in $U^j$. Given that for each $Z_k \in Z$, $Z_k \neq Z$, nodes in $U^{Z_k}$ are direct causes of at least one node in $V_{\mathrm{PA}_j^{\mathcal{G}}}$, any active path between nodes in $U^{Z_k}$ and nodes in $U^j$ would also be an active path between $V_{\mathrm{PA}_j^{\mathcal{G}}}$ and $U^j$, which is a contradiction. Hence $U^j \perp\!\!\!\perp_{\mathcal{G}}^d U^{Z_k}$. $\square$

The previous lemmas allow proving the following result, which will be fundamental to demonstrate the theory of Proposition 3.

**Lemma 4.** *Let $V_j \in V$, and $Z \subset \mathbb{N}$ such that $V_Z = V_{\mathrm{PA}_j^{\mathcal{G}}} \cup \{V_j\}$. Assume $V_{\mathrm{PA}_j^{\mathcal{G}}} \perp\!\!\!\perp_{\mathcal{G}}^d U^j$. W.l.o.g., let the $j$-th element of $V_Z$ be $V_{Z_j} = V_j$. Denote as $U^Z$ the set of unobserved parents of nodes in $V_Z$, and $U^{Z \setminus \{j\}}$ the unobserved parents of nodes in $V_{Z \setminus \{j\}} := V_Z \setminus V_j$. Then, the following holds for each $v_Z, u^Z$ values:*

$$\log p(v_Z) = \log p(v_j | v_{\mathrm{PA}_j^{\mathcal{G}}}) + \log Q(v_Z),$$

*where*

$$Q(v_Z) = \sum_{u^{Z \setminus \{j\}}} p(u^{Z \setminus \{j\}}) \prod_{\substack{k \neq j}}^{|Z|} p(v_{Z_k} | v_{Z_1}, \ldots, v_{Z_{k-1}}, u^{Z_k}).$$

*Proof.* By the law of total probability and the chain rule, we can write $p(v_Z)$ as:

$$\begin{aligned}
p(v_Z) &= \sum_u p(v_Z | u) p(u) \\
&= \sum_u p(u) p(v_{Z_j} | u, v_{Z \setminus \{j\}}) p(v_{Z \setminus \{j\}} | u).
\end{aligned} \tag{20}$$

By Lemma 3, $U^{Z_j} \perp\!\!\!\perp U^{Z_k}$, $k \neq j$, where $U^{Z_k}$ denotes unobserved parents of the node $V_{Z_k}$. Then, we can factorize $p(u) = p\left(u^{Z_j}\right) p\left(u^{Z \setminus \{j\}}\right)$. Plugging the factorization in equation (20) we find

$$\begin{aligned}
p(v_Z) &= \sum_u p\left(u^{Z_j}\right) p\left(u^{Z \setminus \{j\}}\right) p(v_{Z_j} | u, v_{Z \setminus \{j\}}) p(v_{Z \setminus \{j\}} | u) \\
&= \sum_u p\left(u^{Z_j}\right) p\left(u^{Z \setminus \{j\}}\right) p(v_{Z_j} | u^{Z_j}, v_{\mathrm{PA}_{Z_j}^{\mathcal{G}}}) p(v_{Z \setminus \{j\}} | u),
\end{aligned}$$

where the latter equation comes from the global Markov property on the graph $\mathcal{G}$. Further, by assumption of $V_{\mathrm{PA}_j^{\mathcal{G}}} \perp\!\!\!\perp_{\mathcal{G}}^d U^j$, we know that $U^{Z_j} \perp\!\!\!\perp V_{Z_k}$, $k \neq j$, such that $p(v_{Z \setminus \{j\}} | u) = p(v_{Z \setminus \{j\}} | u^{Z \setminus \{j\}})$. Then:

$$\begin{aligned}
p(v_Z) &= \sum_u p\left(u^{Z_j}\right) p\left(u^{Z \setminus \{j\}}\right) p(v_{Z_j} | u^{Z_j}, v_{\mathrm{PA}_{Z_j}^{\mathcal{G}}}) p(v_{Z \setminus \{j\}} | u^{Z \setminus \{j\}}) \\
&= \sum_{u^{Z_j}} p\left(u^{Z_j}\right) p(v_{Z_j} | u^{Z_j}, v_{\mathrm{PA}_{Z_j}^{\mathcal{G}}}) \sum_{u^{Z \setminus \{j\}}} p\left(u^{Z \setminus \{j\}}\right) p(v_{Z \setminus \{j\}} | u^{Z \setminus \{j\}}) \\
&= p(v_{Z_j} | v_{\mathrm{PA}_{Z_j}^{\mathcal{G}}}) \sum_{u^{Z \setminus \{j\}}} p\left(u^{Z \setminus \{j\}}\right) p(v_{Z \setminus \{j\}} | u^{Z \setminus \{j\}}),
\end{aligned}$$

which proves the claim. $\square$

Intuitively, Lemma 4 shows that given a node $V_j$ without children and bidirected edges in a marginalized graph $\mathcal{M}_{V_Z}^{\mathcal{G}}$, the *kernel* of node $V_j$ in the Markov factorization of $p(v_Z)$ is equal to the kernel of the same node in the Markov factorization of $p(x)$ of equation (2), relative to the graph without latent confounders $\mathcal{G}$.

# B   Proofs of theoretical results

## B.1   Proof of Proposition 1

*Proof of Proposition 1.* Observe that

$$\frac{\partial^2}{\partial V_i \partial V_j} \log p(v_Z) = 0 \iff V_i \perp\!\!\!\perp_{\mathcal{G}}^{d} V_j | V_Z \setminus \{V_i, V_j\} \iff V_i \perp\!\!\!\perp_{\mathcal{M}_V^{\mathcal{G}}}^{m} V_j | V_Z \setminus \{V_i, V_j\},$$

where the first equivalence holds by a combination of the faithfulness assumption with the global Markov property, as explicit in equation (3), and the second due to Lemma 1. Then, the claim is proven. □

## B.2   Proof of Proposition 2

*Proof.* The forward direction is immediate from equation (9) and $R_j = N_j$, when $X_j$ is a sink (equation (11)). Thus, we focus on the backward direction. Given

$$\mathbf{E}\left[ \left( \mathbf{E}\left[ \partial_{X_j} \log p(X) \mid R_j = r_j \right] - \partial_{X_j} \log p(X) \right)^2 \right] = 0,$$

we want to show that $X_j$ has no children, which we prove by contradiction.

Let us introduce a function $q : \mathbb{R} \to \mathbb{R}$ such that:

$$\mathbf{E}\left[ \partial_{X_j} \log p(X) \mid R_j = r_j \right] = q(r_j),$$

and $s_j : \mathbb{R}^{|X|} \to \mathbb{R}$,

$$s_j(x) = \partial_{X_j} \log p(x).$$

The mean squared error equal to zero implies that $s_j(X)$ is a constant, once $R_j$ is observed. Formally, under the assumption of $p(x) > 0$ for each $x \in \mathbb{R}^k$, this implies that

$$p(s_j(x) \neq q(R_j) | R_j = r_j) = 0, \forall x \in \mathbb{R}^k.$$

By contradiction, we assume that $X_j$ is not a leaf, and want to show that $s_j(X)$ is not constant in $X$, given $R_j$ fixed. Let $X_i$ such that $X_j \in X_{\mathrm{PA}_i^{\mathcal{G}}}$. Being the structural causal model identifiable, there is no model with distribution $p_X$ whose graph has a backward edge $X_i \to X_j$: thus, the Markov factorization of equation (2) is unique and implies:

$$\partial_{X_j} \log p(X) = \partial_{N_j} \log p(N_j) - \sum_{k \in \mathrm{CH}_j^{\mathcal{G}}} \partial_{X_j} h_k(X_{\mathrm{PA}_k}) \partial_{N_k} \log p(N_k).$$

We note that, by definition of residual in equation (10), $R_j = r_j$ fixes the following distance:

$$R_j = N_j - \mathbf{E}[N_j | X_{\setminus X_j}].$$

Hence, conditioning on $R_j$ doesn't restrict the support of $X$: given $R_j = r_j$, for any $x_{\setminus X_j}$ (value of the vector of elements in $X \setminus \{X_j\}$), $\exists n_j$ with $p(n_j > 0)$ (by the hypothesis of strictly positive densities of the noise terms) that satisfies

$$r_j = n_j - \mathbf{E}[N_j | x_{\setminus X_j}].$$

Next, we condition on all the parents of $X_i$, except for $X_j$, to reduce our problem to the simpler bivariate case. Let $S \subset \mathbb{N}$ and $X_S \subseteq X$ such that $X_{\mathrm{PA}_i^{\mathcal{G}}} \setminus \{X_j\} \subseteq X_S \subseteq X_{\mathrm{ND}_i^{\mathcal{G}}} \setminus \{X_i, X_j\}$, and consider $x_S$ such that $p(x_S > 0)$. Let $X_{\mathrm{PA}_i^{\mathcal{G}}} = x_{\mathrm{PA}_i^{\mathcal{G}}}$ hold under $X_S = x_S$. We define $X_{j_{|x_s}} := X_j | (X_S = x_S)$, and similarly $X_{|x_s} := X | (X_S = x_S)$. Being the SCM a restricted

additive noise model, by Definition 9, the triplet $(g_i, p_{X_{j|x_s}}, p_{N_i})$ satisfies Condition 1, where $g_i(x_j) = h_i(x_{\mathrm{PA}_i^\mathcal{G}\setminus\{X_j\}}, x_j)$. Consider $X_i = x_i$, and the pair of values $(x_j, x_j^*)$ such that $x_j \neq x_j^*$ and

$$\nu''_{N_i}(x_i - g_i(x_j))g'_i(x_j) \neq 0,$$
$$\nu''_{N_i}(x_i - g_i(x_j^*))g'_i(x_j^*) \neq 0,$$

where we resort to the usual notation $\nu_{N_i} := \log p_{N_i}$. By Lemma 2, $(x_i, x_j)$ and $(x_i, x_j^*)$ satisfy:

$$\partial_{X_j}(\xi'(x_j) - \nu'_{N_i}(x_i - g_i(x_j))g'_i(x_j)) \neq 0,$$
$$\partial_{X_j}(\xi'(x_j^*) - \nu'_{N_i}(x_i - g_i(x_j^*))g'_i(x_j^*)) \neq 0,$$

where $\xi := \log p_{X_{j|x_s}}$. Thus, we can fix $x_j$ and $x_j^*$ (which are arbitrarily chosen) such that

$$\partial_{X_j}(\xi'(x_j) - \nu'_{N_i}(x_i - g_i(x_j))g'_i(x_j)) - \partial_{X_j}(\xi'(x_j^*) - \nu'_{N_i}(x_i - g_i(x_j^*))g'_i(x_j^*)) \neq 0. \quad (21)$$

Fixing $X_{|x_S, x_j} = x$ and $X_{|x_S, x_j^*} = x^*$, where the two values differ only in their j-$th$ component, we find the following difference:

$$s_j(x) - s_j(x^*) = \partial_{X_j}(\xi'(x_j) - \nu'_{N_i}(x_i - g_i(x_j))g'_i(x_j)) - \partial_{X_j}(\xi'(x_j^*) - \nu'_{N_i}(x_i - g_i(x_j^*))g'_i(x_j^*)),$$

which is different from 0 by equation (21). This contradicts the fact that the score $s_j$ is constant once $R_j$ is fixed, which proves our claim. $\qquad \square$

### B.3   Proof of Proposition 3

In this proof, we use several ideas from the demonstration of Proposition 2. We demonstrate the forward and the backward parts of the two statements separately.

*Proof of part (i), forward direction.* Given $V_Z = V_{\mathrm{PA}_j^\mathcal{G}} \cup \{V_i, V_j\}$ and $r_j \in \mathbb{R}$ in the image of $R_j$, we want to show:

$$V_{\mathrm{PA}_j^\mathcal{G}} \perp\!\!\!\perp_\mathcal{G}^d U^j \wedge V_i \in V_{\mathrm{PA}_j^\mathcal{G}} \implies \mathbf{E}[\partial_{V_j} \log p(V_Z) - \mathbf{E}[\partial_{V_j} \log p(V_Z)|R_j(V_Z) = r_j]]^2 = 0.$$

By Lemma 4, the score of $V_j$ is

$$\partial_{V_j} \log p(V_Z) = \partial_{V_j} \log p(V_j|V_{\mathrm{PA}_j^\mathcal{G}}) + \partial_{V_j} \log Q(V_Z)$$
$$= \log p(\tilde{N}_j),$$

for some $Q$ map acting on $V_Z$. The latter equality holds because all variables in $V_Z$ are non-descendants of $V_j$, such that $\partial_{V_j} Q(V_Z) = 0$. Further, by equation (16) we know that

$$R_j(V_Z) = \tilde{N}_j + c,$$

where $c = -\mathbf{E}[\tilde{N}_j]$ is a constant. It follows that the least square estimator of the score of $V_j$ from $R_j(V_Z)$ satisfies the following equation:

$$\mathbf{E}[\partial_{V_j} \log p(V_Z)|R_j(V_Z)] = \mathbf{E}[\partial_{V_j} \log p(\tilde{N}_j)|\tilde{N}_j] = \partial_{V_j} \log p(\tilde{N}_j),$$

where the first equality holds because $\mathbf{E}[\cdot|\tilde{N}_j] = \mathbf{E}[\cdot|\tilde{N}_j + c]$. Then, we find

$$\mathbf{E}[\partial_{V_j} \log p(V_Z) - \mathbf{E}[\partial_{V_j} \log p(V_Z)|R_j(V_Z) = r_j]]^2 = \mathbf{E}[\partial_{V_j} \log p(\tilde{N}_j) - \partial_{V_j} \log p(\tilde{N}_j)]^2 = 0,$$

which is exactly our claim. $\qquad \square$

*Proof of part (i), backward direction.* Given $V_Z = V_{\mathrm{PA}_j^\mathcal{G}} \cup \{V_i, V_j\}$, $r_j \in \mathbb{R}$ in the image of $R_j$, and

$$\mathbf{E}[\partial_{V_j} \log p(V_Z) - \mathbf{E}[\partial_{V_j} \log p(V_Z)|R_j(V_Z) = r_j]]^2 = 0, \quad (22)$$

we want to show that $V_{\mathrm{PA}_j^\mathcal{G}} \perp\!\!\!\perp_\mathcal{G}^d U^j \wedge V_i \in V_{\mathrm{PA}_j^\mathcal{G}}$, meaning that there is a direct causal effect that is not biased by unobserved variables. We provide the proof by contradiction, in analogy to the demonstration of the backward direction of Proposition 2.

Let us introduce $s_j : \mathbb{R}^{|V_Z|} \to \mathbb{R}$,

$$s_j(v_Z) = \partial_{V_j} \log p(V_Z).$$

The mean squared error equal to zero implies that $s_j(V_Z)$ is constant in $V_Z$, once $R_j$ is observed. By contradiction, we assume that $V_{\mathrm{PA}_j^{\mathcal{G}}} \not\perp\!\!\!\perp_{\mathcal{G}}^d U^j \vee V_i \notin V_{\mathrm{PA}_j^{\mathcal{G}}}$, and want to show that $s_j(V_Z)$ is not constant in $V_Z$, given $R_j$ fixed. In this regard, we make the following observation: by definition of residual in equation (15), $R_i(V_Z) = r_i$ fixes the following distance:

$$R_j(V_Z) = \tilde{N}_j - \mathbf{E}[\tilde{N}_j | V_{Z \setminus \{j\}}].$$

Hence, conditioning on $R_j(V_Z)$ doesn't restrict the support of $V_Z$: given $R_j(V_Z) = r_j$, $\exists \tilde{n}_j$ with $p(\tilde{n}_j) > 0$ (by assumption of strictly positive densities $p_{N_j}$ and $p_X$), that satisfies

$$r_j = \tilde{n}_j - \mathbf{E}[\tilde{N}_j | v_{Z \setminus \{j\}}],$$

for all $v_{Z \setminus \{j\}}$. Hence, the random variable $V_Z | R_j(V_Z) = r_j$ has strictly positive density on all points $v_Z$ where $p_{V_Z}(v_Z) > 0$. Now, consider $v_Z$ and $v_Z^*$, taken from the set of uncountable values such that the score $s_j$ function is not a constant, meaning that $s_j(v_Z) \neq s_j(v_Z^*)$, where $V_Z$ is sampled given $R_j(V_Z) = r_j$. Given that different $v_Z$ and $v_Z^*$ are selected from an uncountable subset of the support, we conclude that the score $s_j | (R_j(V_Z) = r_j) = \partial_{V_j} \log p(V_Z | R_j(V_Z) = r_j)$ is not a constant for at least an uncountable set of points, which contradicts equation (22). $\qquad\square$

*Proof of part (ii), forward direction.* Given that $V_i$ is connected to $V_j$ in the marginal MAG and that $V_{\mathrm{PA}_j^{\mathcal{G}}} \not\perp\!\!\!\perp_{\mathcal{G}}^d U^j \vee V_i \notin V_{\mathrm{PA}_j^{\mathcal{G}}}$, we want to show that for each $V_Z \subseteq V$ with $\{V_i, V_j\} \subseteq V_Z$, the following holds:

$$\mathbf{E}[\partial_{V_j} \log p(V_Z) - \mathbf{E}[\partial_{V_j} \log p(V_Z) | R_j(V_Z) = r_j]]^2 \neq 0. \tag{23}$$

Let us introduce $h : \mathbb{R} \to \mathbb{R}$ such that:

$$\mathbf{E}[\partial_{V_j} \log p(V_Z) | R_j(V_Z) = r_j] = h(r_j),$$

and further define:

$$s_j(V_Z) = \partial_{V_j} \log p(V_Z).$$

Having the mean squared error in equation (23) equals zero implies that $s_j(V_Z)$ is a constant, once $R_j(V_Z)$ is observed. Thus, the goal of the proof is to show that there are values of $V_Z$ such that the score is not a constant once $R_j$ is fixed. By definition of residual in equation (15), $R_j(V_Z) = r_j$ fixes the following distance:

$$R_j(V_Z) = \tilde{N}_j - \mathbf{E}[\tilde{N}_j | V_{Z \setminus \{j\}}].$$

Hence, conditioning on $R_j(V_Z)$ doesn't restrict the support of $V_Z$: given $R_j(V_Z) = r_j$, $\exists \tilde{n}_j$ with $p(\tilde{n}_j) > 0$ (by assumption of positive density of the noise $N_j$ on the support $\mathbb{R}$), that satisfies

$$r_j = \tilde{n}_j - \mathbf{E}[\tilde{N}_j | v_{Z \setminus \{j\}}],$$

for all $v_{Z \setminus \{j\}}$. Hence, the random variable $V_Z | R_j(V_Z) = r_j$ has strictly positive density on all points $v_Z$ where $p_{V_Z}(v_Z) > 0$. Now, consider $v_Z$ and $v_Z^*$, taken from the set of uncountable values such that the score $s_j$ function is not a constant, meaning that $s_j(v_Z) \neq s_j(v_Z^*)$, where $V_Z$ is sampled given $R_j(V_Z) = r_j$. Given that different $v_Z$ and $v_Z^*$ are selected from an uncountable subset of the support, we conclude that the score $s_j | (R_j(V_Z) = r_j) = \partial_{V_j} \log p(V_Z | R_j(V_Z) = r_j)$ is not a constant for at least an uncountable set of points, such that the claim follows. $\qquad\square$

*Proof of part (ii), backward direction.* Given that $\mathbf{E}[\partial_{V_j} \log p(V_Z) - \mathbf{E}[\partial_{V_j} \log p(V_Z) | R_j(V_Z) = r_j]]^2 \neq 0$ for all $V_Z \subseteq V$ such that $\{V_i, V_j\} \in V_Z$, and given $V_i$ and $V_j$ adjacent in the marginal MAG, we want to show that

$$V_{\mathrm{PA}_j^{\mathcal{G}}} \not\perp\!\!\!\perp_{\mathcal{G}}^d U^j \vee V_i \notin V_{\mathrm{PA}_j^{\mathcal{G}}}.$$

The prove comes easily by contradiction: say that $V_{\mathrm{PA}_j^{\mathcal{G}}} \perp\!\!\!\perp_{\mathcal{G}}^d U^j \wedge V_i \in V_{\mathrm{PA}_j^{\mathcal{G}}}$. Then, by the forward direction of part *(i)* of Proposition 3, we know that $V_Z = V_{\mathrm{PA}_j^{\mathcal{G}}} \cup \{V_j\}$ satisfies $\mathbf{E}[\partial_{V_j} \log p(V_Z) - \mathbf{E}[\partial_{V_j} \log p(V_Z) | R_j(V_Z) = r_j]]^2 = 0$, leading to a contradiction. $\qquad\square$

## C  Algorithm

### C.1  Detailed description of our algorithm

In Proposition 1 we have seen that score matching can detect $m$-separations and therefore the skeleton of the PAG describing the data. If one is willing to make the assumptions required for Proposition 3 it could be desirable to use this to orient edges, since the interpretation of PAG edges might be cumbersome for people not familiar with ancestral models. Therefore, one could simply find the skeleton of the PAG using the fast adjacency search [5] and then orient the edges by applying Proposition 3 on every subset of the neighbourhood of every node. This would yield a very costly algorithm. But if we make the assumptions required to orient edges with Proposition 3 we can do a bit better. In Algorithm 2 we present an algorithm that still has the same worst case runtime but runs polynomially in the best case. The main intuition is that we iteratively remove irrelevant nodes in the spirit of the original SCORE algorithm [15]. To this end, we first check if the is any unconfounded sink if we consider the set of all remaining variables. If there is one, we can orient its parents and ignore it afterwards. If there is no such set, we need to fall back to the procedure proposed above, i.e. we need to check the condition of Proposition 3 on all subsets of the neighbourhood of a node, until we find no node with a direct outgoing edge. In Proposition 4 we show that this way we do not fail orient edge or fail to remove any adjacency. In the following discussion, we will use the notation

$$\delta_i(X_Z) := \mathbf{E}[\partial_{V_j} \log p(V_Z) - \mathbf{E}[\partial_{V_j} \log p(V_Z)|R_j(V_Z) = r_j]]^2,$$

for the second residual from Proposition 3 and also

$$\delta_{i,j}(X_Z) := \frac{\partial^2}{\partial V_i \partial V_j} \log p(v_Z)$$

for the cross-partial derivative, where $X_i, X_j \in V$ and $Z \subseteq V$.

**Proposition 4** (Correctness of algorithm). *Let $X = V \dot{\cup} U$ be generated by the SCM in Equation (4) with non-constant scores for uncountably many values. Let $\mathcal{G}_X$ be the causal DAG of $X$ and $\mathcal{G}_V$ be the marginal MAG of $\mathcal{G}_X$. Then Algorithm 2 outputs a directed edge from $X_i \in V$ to $X_j \in V$ iff there is a direct edge in $\mathcal{G}_X$ between them and no unobserved backdoor path w.r.t. $U$. Further, the output of Algorithm 2 has the same skeleton as $\mathcal{G}_V$.*

*Proof.* We proof the statement by induction over the steps of the algorithm. Let $S$ be the set of remaining nodes in an arbitrary step of the algorithm. Our induction hypothesis is that for $X_i, X_j \in S$ and $X_k \in B_i$ we have

1. $X_i$ is an unconfounded sink w.r.t. to some set $S' \subseteq S$ iff $X_i$ is an unconfounded sink w.r.t. some $S'' \subseteq V$

2. if there is no $S' \subseteq V \setminus \{X_i, X_j\}$ such that $X_i \perp\!\!\!\perp X_j \mid S'$ then $X_j \in B_i$

Clearly, this holds in the initial step as $S = V$.

Suppose we find $\delta_i(X_S) = 0$ for $X_i \in S$. If $X_i$ has at least one adjacent node in $\mathcal{M}_V^{\mathcal{G}}$, by Proposition 3, we know that $X_i$ does not have any children and is also not connected to any other node in $S$ via a hidden mediator or unobserved confounder. This means, all nodes that are not separable from $X_i$ must be direct parents of $X_i$, which are by our induction hypothesis 2) the nodes in $B_i$. Since $X_i$ does not have children, it also suffices to check $X_i \perp\!\!\!\perp X_j|S \setminus \{X_i, X_j\}$ for $X_j \in B_i$ (instead of conditioning on all subsets of $B_i$). So we can already add these direct edges to the output. If, on the other hand, $X_i$ has no adjacent nodes in $\mathcal{M}_V^{\mathcal{G}}$, we have $X_i \perp\!\!\!\perp X_j|S \setminus \{X_i, X_j\}$ for $X_j \in B_i$, so in both cases we add the correct set of parents. Since $X_i$ is not an ancestor of any of the nodes in $S \setminus \{X_i\}$, $X_i$ cannot be a hidden mediator or hidden confounder between nodes in $S \setminus \{X_i\}$ and conditioning on $X_i$ cannot block an open path. Thus, the induction hypothesis still holds in the next step.

Suppose now there is no unconfounded sink and we explore $X_i$. By our induction hypothesis 2), $B_i$ contains the parents of $X_i$ and by Proposition 3 it suffices to only look at subsets of $B_i$ to orient direct edges. And also due to the induction hypothesis 2) $B_i$ contains all nodes that are not separable from $X_i$. So by adding bidirected edges to all nodes in $B_i$ can only add too many edges but not miss some.

---

**Algorithm 2** AdaScore Algorithm

---

**procedure** ADASCORE($p, X_1, \ldots, X_d$)
    $S \leftarrow \{X_1, \ldots, X_d\}$                                                 ▷ Remaining nodes
    $E \leftarrow \{\ \}$                                                       ▷ Edges
    **for** $X_i \in S$ **do**
        $B_i \leftarrow \{X_1, \ldots, X_d\}$                                   ▷ Neighbourhoods
    **while** $S \neq \emptyset$ **do**                             ▷ While nodes remain
        **if** $\exists X_i \in S : \delta_i(\mathbf{X}_S) = 0$ **then**         ▷ If there is an unconfounded sink
            $S \leftarrow S \setminus \{X_i\}$
            $E \leftarrow E \cup \{X_j \rightarrow X_i : \delta_{i,j}(\mathbf{X}_S) \neq 0\}$     ▷ Add edges like DAS
        **else**
            **for** $X_i \in S$ **do**
                **for** $X_j \in B_i$ **do**                   ▷ Prune neighbourhoods
                    **if** $\delta_{i,j}(\mathbf{X}_S) = 0$ **then**
                        $B_i \leftarrow B_i \setminus \{X_j\}$
                        $B_j \leftarrow B_j \setminus \{X_i\}$
                **for** $X_j \in B_i$ **do**                   ▷ Orient edges in $B_i$
                    $m_i = \min_{S' \subseteq B_i} \delta_i(\mathbf{X}_{S' \cup \{X_i\}})$
                    $m_j = \min_{S' \subseteq B_j} \delta_j(\mathbf{X}_{S' \cup \{X_j\}}))$
                    **if** $m_i = 0 \wedge m_j \neq 0$ **then**
                        $E \leftarrow E \cup \{X_j \rightarrow X_i\}$
                    **else if** $m_i \neq 0 \wedge m_j = 0$ **then**
                        $E \leftarrow E \cup \{X_i \rightarrow X_j\}$
                    **else**
                        $E \leftarrow E \cup \{X_i \leftrightarrow X_j\}$
                **if** $\exists X_j \in B_i : (X_i \rightarrow X_j) \in E$ **then**
                    **continue with** $X_j$
                **else**                 ▷ $X_i$ has no unconfounded outgoing edge
                    $S \leftarrow S \setminus \{X_i\}$                 ▷ Remove $X_i$
                    **break**
        **for** $X_i \leftrightarrow X_j \in E$ **do**                  ▷ Prune bidirected edges
            **if** $\min_{S' \subseteq \mathrm{Adj}(X_i)} \delta_{i,j}(\mathbf{X}_{S' \cup \{X_i\}}) = 0 \vee \min_{S' \subseteq \mathrm{Adj}(X_j)} \delta_{i,j}(\mathbf{X}_{S' \cup \{X_i\}}) = 0$ **then**
                $E \leftarrow E \setminus \{X_i \leftrightarrow X_j\}$
    **return** $E$

---

Now it remains to show that the induction hypothesis holds if we set $S$ to $S \setminus \{X_i\}$. For 1) we need to show that $X_i$ cannot be a hidden mediator or hidden confounder w.r.t. $S \setminus \{X_i\}$ (since ignoring $X_i$ won't change whether there is a direct edge or not). Suppose $X_i$ is on a unobserved causal path $X_k \rightarrow \cdots \rightarrow U^m \rightarrow X_l$ with $X_k, X_l \in S \setminus \{X_i\}$ and $U^m \in X \setminus (S \setminus \{X_i\})$. This path must have been a unobserved causal path before, unless $X_i = U^m$. But then there is a direct edge $X_i \rightarrow X_l$. We would not remove $X_i$ from $S$ if this edge was unconfounded, so there must a hidden confounder between $X_i$ and $X_l$. But in this case, Proposition 3 wouldn't allow us to direct the edge anyway, since $V_{\mathrm{PA}_l} \not\perp^d_{\mathcal{G}} U_l$. Suppose there is confounding path $X_k \leftarrow \cdots \rightarrow U^m \rightarrow X_l$ with $X_k, X_l \in S \setminus \{X_i\}$ and $U^m \in X \setminus (S \setminus \{X_i\})$. If $X_i \neq U^m$ the path was already been a confounding path without $X_i$ being unobserved. So again, there must be a confounder between $X_i$ and $X_l$, as otherwise we would not remove $X_i$. And analogously to before, we could not have oriented the edge even with $X_i \in S$ since $V_{\mathrm{PA}_l} \not\perp^d_{\mathcal{G}} U_l$. For 2) we only have to see that we just remove nodes from $B_i$ if we found an independence.

For $|S| < 2$, the algorithm enters the final pruning stage. From the discussion above it is clear, that we already have the correct result, up to potentially too many bidirected edges. In the final step we certainly remove all these edges $X_i \leftrightarrow X_j$, as we check $m$-separation for all subsets of the neighbourhoods $\mathrm{Adj}(X_i)$ and $\mathrm{Adj}(X_j)$, which are supersets of the true neighbourhoods.

$\square$

## C.2 Finite sample version of AdaScore

All theoretical results in the paper have assumed that we know the density of our data. Obviously, in practise we have to deal with a finite sample instead. Especially, in Proposition 1 and Proposition 3 we derived criteria that compare random variables with zero. Clearly, this condition is never met in practise. Therefore, we need find ways to reasonably set thresholds for these random quantities.

First note, that we use the Stein gradient estimator [35] to estimate the score function. This means especially that for a node $V_i$ we get a vector

$$\left( \left( \frac{\partial}{\partial V_i} \log p(v) \right)_l \right)_{l=1,\ldots,m}, \tag{24}$$

i.e. an estimate of the score for every one of the $m$ samples. Analogously, we get a $m \times d \times d$ tensor for the estimates of $\frac{\partial^2}{\partial V_i \partial V_j} \log p(v)$.

In Proposition 1 we showed that

$$\frac{\partial^2}{\partial V_i \partial V_j} \log p(v_Z) = 0 \iff X_i \perp\!\!\!\perp_{\mathcal{M}_V^{\mathcal{G}}}^m V_j | V_Z \setminus \{V_i, V_j\}.$$

In the finite sample version, we use a one sample t-test on the vector of estimated cross-partial derivatives with the null-hypothesis that the means is zero. Due to the central limit theorem, the sample mean follows approximately a Gaussian distribution, regardless of the true distribution of the observations.

For Proposition 3 we need to do some additional steps. Recall, that the relevant quantity in Proposition 3 is the mean squared error of a regression, which is always positive. Therefore, a test for mean zero is highly likely to reject in any case. We decided to employ a two-sample test in a similar (but different) manner as Montagna et al. [17]. As test, we used the Mann-Whitney U-test. Note, that Algorithm 2 employs Proposition 3 in two different ways: first, to decide whether there is an unconfounded sink and second, to orient edges in case there is no unconfounded sink. We pick a different sample as second sample of the Mann-Whitney U-test.

Analogously to before, this is a vector with $m$ entries, one for every sample.

Note, that in the case where we want to check if there is an unconfounded sink, we do not make any mistake by rejecting too few hypotheses, i.e. if we miss some unconfounded sinks (instead, we only lose efficiency, as we do the costly iteration over all possible sets of parents). Therefore, for this test we chose a a second sample that yields a "conservative" test result.

As candidate sink for set $S \subseteq V$, we pick the node $X_i = \min_i \text{mean}(\delta_i(X_S))$. In fact, we want to know whether the mean of $\delta_i$ is significantly lower than *all* other means. But we empirically observed that choosing the concatenated $\delta$s of all nodes as second sample makes the test reject with very high probability, which would lead our algorithm to falsely assume the existence of an unconfoudned sink. Instead, we then pick as second "reference node" $X_j = \min_{j \neq i} \text{mean}(\delta_j(X_Z))$. We then do the two sample test between $\delta_i(X_Z)$ and $\delta_j(X_Z)$. The intuition is that the test will reject the hypothesis of identical means, if $X_i$ is an unconfounded sink but $X_j$ is not.

In the case where we use Proposition 3 to orient edges, we only need to decide whether an not previsouly directed edge $X_i - X_j$ needs to be oriented one way, the other way, or not at all. Instead, here the issue lies in the fact that we need to iterate over possible sets of parents of the nodes. Let $B_i$ be the set of nodes that have not been $m$-separated from $X_i$ by any test so far. We pick the subset $Z_i = \min_{Z' \subseteq B_i} \text{mean}(\delta_i^{Z'})$, i.e. the set with the lowest mean error. We then conduct the test with $\delta_i(X_{Z_i})$ and $\delta_j(X_{Z_j})$. If there is a directed edge between them, one of the residuals will be significantly lower than the other.

Just like Montagna et al. [17] we use a cross-validation scheme to generate the residuals, in order to prevent overfitting. We split the dataset into several equally sized, disjoint subsamples. For every residual we fit the regression on all subsamples that don't contain the respective target.

Also, just like in the NoGAM algorithm Montagna et al. [17] we add a pruning step for the directed edges to the end. The idea is to use a feature selection method to remove insignificant edges. Just like Montagna et al. [17], we use the CAM-based pruning step proposed by Bühlmann et al. [36], which fits a generalised additive regression model from the parents to a child and test whether one of the

additive components is significantly non-zero. All parents for which the test rejects this hypothesis are removed.

## C.3 Complexity

**Proposition 5.** *Complexity Let $n$ be the number of samples and $d$ the number of observable nodes. Algorithm 2 runs in*

$$\Omega\left((d^2 - d) \cdot (r(n,d) + s(n,d))\right) \quad and \quad \mathcal{O}\left(d^2 \cdot 2^d(r(n,d) + s(n,d))\right),$$

*where $r(n,d)$ is the time required to solve a regression problem and $s(n,d)$ is the time for calculating the score. With e.g. kernel-ridge regression and the Stein-estimator, both run in $\mathcal{O}(n^3)$.*

*Proof.* Algorithm 2 runs its main loop $d$ times. It first checks for the existence of an unconfounded sink, which involves solving $2d$ regression problems (including cross-validation prediction) and calculating the score, adding up to $(2d^2 - d)$ regressions and $d$ score evaluations. In the worst case, we detect no unconfounded sink and iterate through all subsets of the neighbourhood of a node (which is in the worst case of size $d - 1$) and for all other nodes in the neighbourhood we solve $2d$ regression problems and evaluate the score. For each subset we calculate two regression functions, the score and calculate the entries in the Hessian of the log-density, i.e. $d \cdot 2^d$ regressions, $d \cdot 2^{d-1}$ scores and additionally $2^{d-1}$ Hessians. If we are unlucky, this node has a directed outgoing edge and we continue with this node (with the same size of nodes). This can happen $d - 1$ times. So we get $(d^2 - d) \cdot 2^d$ regressions and $(d^2 - d) \cdot 2^{d-1}$ scores and Hessians. In the final pruning step we calculate for every bidirected edge (of which there can be $(d^2 - d)/2$) a Hessian for all subsets of the neighbourhoods, which can again be $2^{d-1}$ subsets. Using the pruning procedure from CAM for the directed edges we also spend at most $\mathcal{O}(nd^3)$ steps.

In the best case, we always find an unconfounded sink. Then our algorithm reduces to NoGAM.

$\square$

# D Experimental details

In this section, we present the details of our experiments in terms of synthetic data generation and algorithms hyperparameters.

## D.1 Synthetic data generation

In this work, we rely on synthetic data to benchmark AdaScore's finite samples performance. For each dataset, we first sample the ground truth graph and then generate the observations according to the causal graph.

**Erdös-Renyi graphs.** The ground truth graphs are generated according to the Erdös-Renyi model. It allows specifying the number of nodes and the probability of connecting each pair of nodes). In ER graphs, a pair of nodes has the same probability of being connected.

**Nonlinear causal mechanisms.** Nonlinear causal mechanisms are parametrized by a neural network with random weights. We create a fully connected neural network with one hidden layer with 10 units, Parametric ReLU activation function, followed by one normalizing layer before the final fully connected layer. The weights of the neural network are sampled from a standard Gaussian distribution. This strategy for synthetic data generation is commonly adopted in the literature [26, 18, 28, 29, 27].

**Linear causal mechanisms.** For the linear mechanisms, we define a simple linear regression model predicting the effects from their causes and noise terms, weighted by randomly sampled coefficients. Coefficients are generated as samples from a Uniform distribution supported in the range $[-3, -0.5] \cup [0.5, 3]$. We avoid too small coefficients to avoid *close to unfaithful* datasets Uhler et al. [24].

**Noise terms distribution.** The noise terms are sampled from a Uniform distribution supported between $-2$ and $2$.

Finally, we remark that we standardize the data by their empirical data. This is known to remove shortcuts that allow finding a correct causal order sorting variables by their marginal variance, as in *varsortability*, described in Reisach et al. [32], or sorting variables by the magnitude of their score $|\partial_{X_i} \log p(X)|$, a phenomenon known as *scoresortability* analyzed by Montagna et al. [18].

## D.2 AdaScore hyperparameters

For AdaScore, we set the $\alpha$ level for the required hypothesis testing at $0.05$. For the CAM-pruning step, the level is instead set at $0.001$, the default value of the dodidscover Python implementation of the method, and commonly found in all papers using CAM-pruning for edge selection [15, 16, 17, 36]. For the remaining parameters. The regression hyperparameters for the estimation of the residuals are found via cross-validation during inference: tuning is done minimizing the generalization error on the estimated residuals, without using the performance on the causal graph ground truth. Finally, for the score matching estimation, the regularization coefficients are set to $0.001$.

## D.3 Computer resources

All experiments have been run on an AWS EC2 instance of type `p3.2xlarge`. These machines contain Intel Xeon E5-2686-v4 processors with 2.3 GHz and 8 virtual cores as well as 61 GB RAM. All experiments can be run within a day.

## E  Additional Experiments

In this section, we provide additional experimental results. All synthetic data has been generated as described in Appendix D.1.

### E.1  Non-additive mechanisms

In Figure 1 we have demonstrated the performance of our proposed method on data generated by linear SCMs and non-linear SCMs with additive noise. But Proposition 1 also holds for *any* faithful distribution generated by an acyclic model. Thus, we employed as mechanism a neural network-based approach similar to the non-linear mechanism described in Appendix D. Instead of adding the noise term, we feed it as additional input into the neural network. Results in this setting are reported in Figure 2. As neither AdaScore nor any of the baseline algorithms has theoretical guarantees for the orientation of edges in this scenario, we report the $F_1$-score (popular in classification problems) w.r.t. to the existence of an edge, regardless of orientation. Our experiments show that AdaScore can, in general, correctly recover the graph's skeleton in all the scenarios, with an $F_1$ score median between 1 and $\sim 0.75$, respectively for small and large numbers of nodes.

### E.2  Sparse graphs

In this section, we present the experiments on sparse Erdös-Renyi graphs where each pair of nodes is connected by an edge with probability $0.3$. The results are illustrated in Figure 3. For sparse graphs, recovery results are similar to the dense case, with AdaScore generally providing comparable performance to the other methods.

### E.3  Increasing number of samples

In the following series of plots we demonstrate the scaling behaviour of our method w.r.t. to the number of samples. Figure 5 shows results with edge probability 0.5 and Figure 4 with 0.3. All graphs contain seven observable nodes. As before we observe that AdaScore performs comparably to other methods. E.g. in Figures 4a and 5b we can see that the median error AdaScore improves with additional samples and in all plots we see that no other algorithm seems to gain an advantage over AdaScore with increasing sample size.

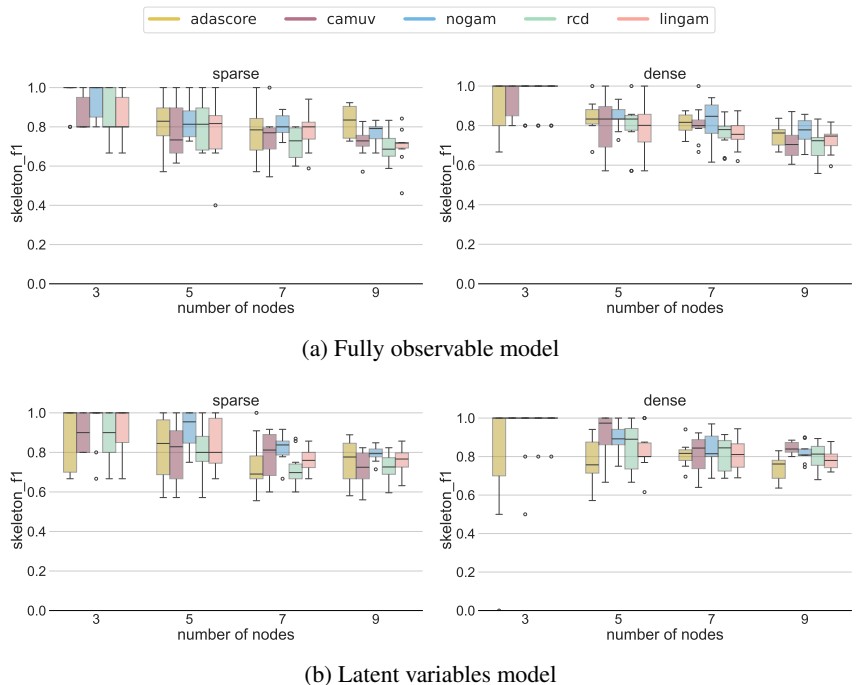

(a) Fully observable model

(b) Latent variables model

Figure 2: Empirical results for non-additive causal mechanisms on sparse graphs with different numbers of nodes, on fully observable (no hidden variables) and latent variable models. We report the $F_1$ score w.r.t. the existence of edges (the higher, the better).

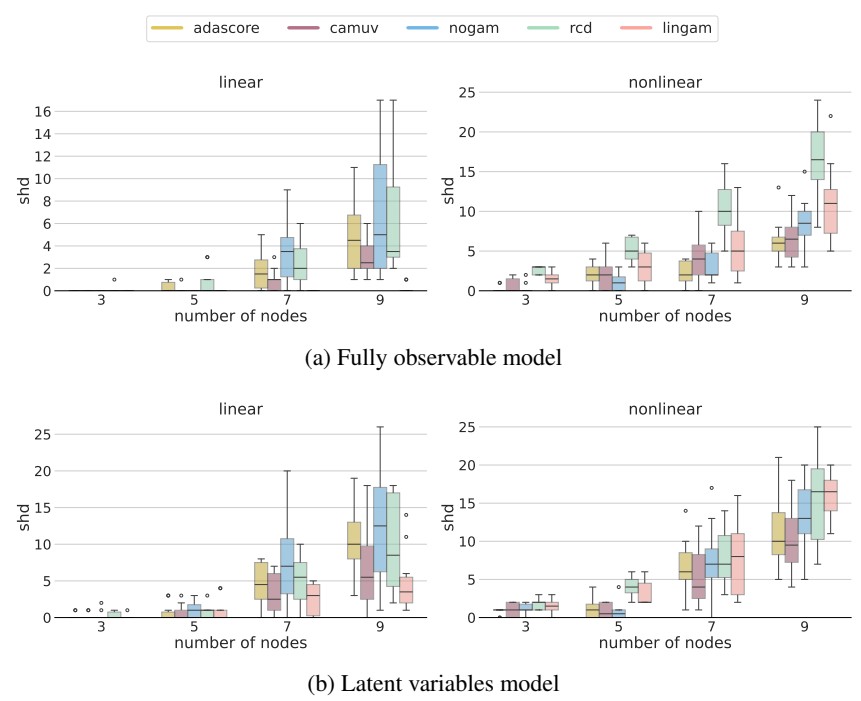

(a) Fully observable model

(b) Latent variables model

Figure 3: Empirical results on sparse graphs with different numbers of nodes, on fully observable (no hidden variables) and latent variable models. We report the SHD accuracy (the lower, the better).

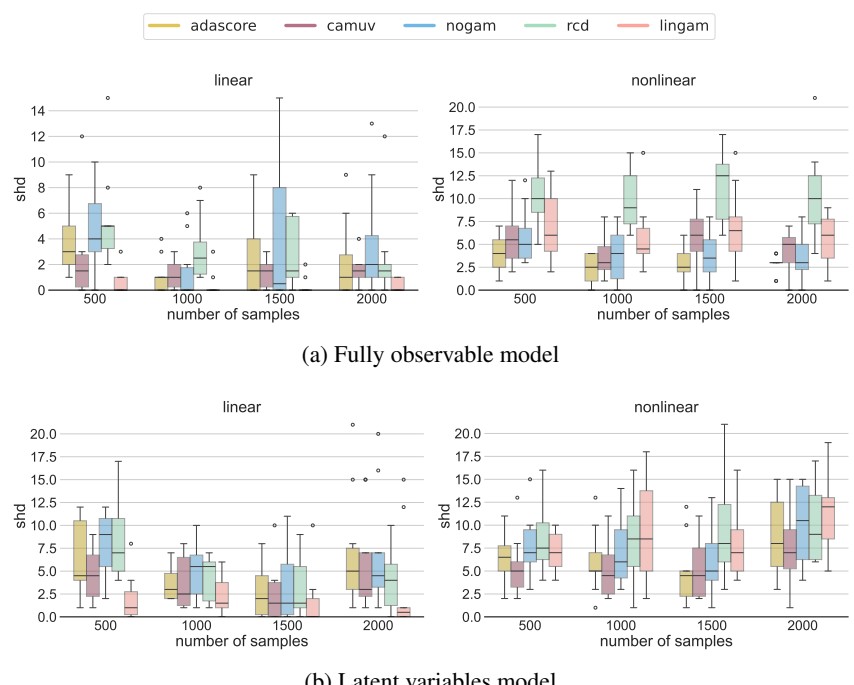

(a) Fully observable model

(b) Latent variables model

Figure 4: Empirical results on sparse graphs with different numbers of samples and seven nodes, on fully observable (no hidden variables) and latent variable models. We report the SHD accuracy (the lower, the better).

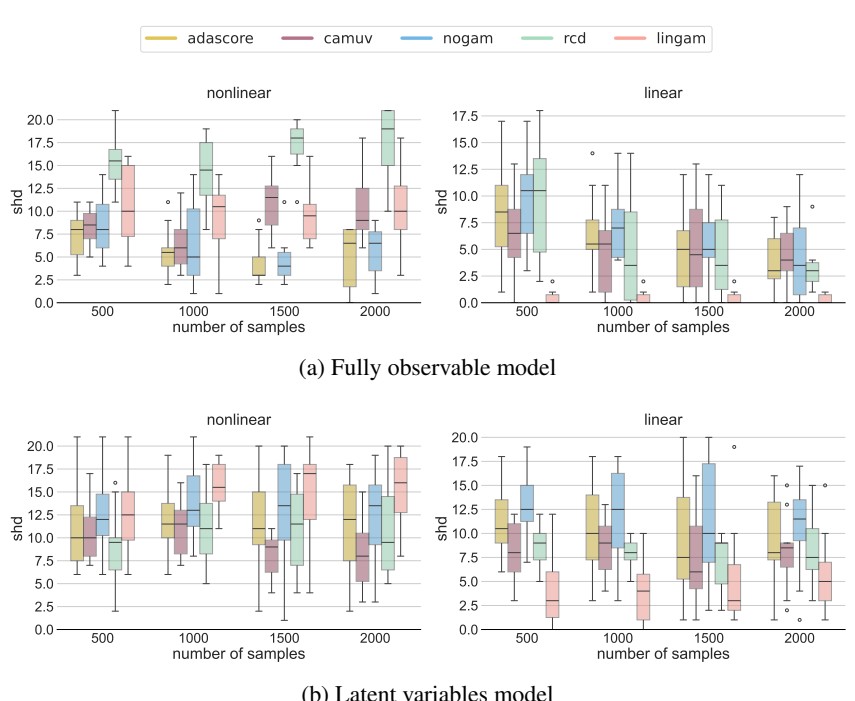

(a) Fully observable model

(b) Latent variables model

Figure 5: Empirical results on dense graphs with different numbers of samples and seven nodes, on fully observable (no hidden variables) and latent variable models. We report the SHD accuracy (the lower, the better).

### E.4 Limitations

In this section, we remark the limitations of our empirical study. It is well known that causal discovery lacks meaningful, multivariate benchmark datasets with known ground truth. For this reason, it is common to rely on synthetically generated datasets. We believe that results on synthetic graphs should be taken with care, as there is no strong reason to believe that they should mirror the benchmarked algorithms' behaviors in real-world settings, where often there is no prior knowledge about the structural causal model underlying available observations.

