# OpenReview forum: "Score matching through the roof: linear, nonlinear, and latent variables causal discovery"
_NeurIPS.cc/2024/Conference — Submitted to NeurIPS 2024_

### Official Review · Reviewer_WsKx · 2024-06-20

**Soundness:** 4
**Presentation:** 3
**Contribution:** 4
**Rating:** 8
**Confidence:** 4

**Summary:**

This paper present novel theoretical results to identify causal effects in restricted ANMs even in case of unobserved confounders.

**Strengths:**

**The paper provides novel contributions to the field of score-based causal discovery by extending previous works to confounded restricted ANMs. Based on these contributions, I strongly support this paper's acceptance.**

- the problem definition is very useful to orient readers
- the theoretical results are novel
- the experiments compare against a sufficient number of baselines, and though the proposed method is not SOTA, it compares well and has better theoretical guarantees


Based on the authors' response, I am eager to improve my score.

**Weaknesses:**

I have a few remarks on improving the flow of the paper; however, even the first four points are not considered major issues.

- Even though condition 1 is a well-known result in the causality literature, I suggest explaining why that admits linear models and including some description of **restricted ANMs** in the main text (at least for me, it is not evident, especially since the condition lacks intuition). To be clear, even this point does not diminish the main contribution, which I see regarding the results for confounders.
- As **inducing paths** are an important concept for the main contributions, please _include it in the main text_ if space permits (suggestion: you can reduce spacing in $\texttt{itemize}$ by setting $\texttt{\\\\begin\\{itemize\\}[nolistsep]}$ )
- I could **not find the definition of an active path** (not even in Def. 5, where it is said to be defined); I presume it is a path that is not blocked, but it would be better to state this explicitly. Maybe it would even be better to use "a path that is not blocked" instead of introducing new terminology (this is the first time I encountered "active paths"; I could be wrong about this)
- The text is sometimes difficult to follow, due to heavy reliance on notation. I'd consider delegating the not crucial part to the appendix (potential candidates in 2.1) and using the remaining space to explain the main quantities better, especially the residuals (e.g., Eq. 12)


## Minor points
- please specify what you are calculating the expectation with respect to (using a bold E is also unconventional, though it's clear from the context it's an expectation)
- as the mathematical objects for d-/m-separation are distinguishable, you might consider dropping the superscript to simplify notation; also, I'd suggest adding whitespace after $\perp^d_\mathcal{G}$ and the like to make it easier for the reader to attribute the indices to $\perp$ and not the not on its right
- I could not find the definition for $\dot{\cup}$
- in the explanation of Prop 3., the wording makes it a bit hard to discern that you also provide intuition for the second part; it would help if you refer to _Part (ii)_ explicitly
- _Score matching through the roof_ in the title does not have added value for me, I'd consider rephrasing it to convey the message that "we propose score-based causal discovery methods for confounded restricted ANMs"

**Questions:**

Some methods in the literature [1-4] use the Jacobian of a learned neural network instead of relying on the score for causal discovery/identifiability claims. How do you see the relation of your contribution to those Jacobian-based methods?


- [1] Sébastien Lachapelle, Philippe Brouillard, Tristan Deleu, and Simon Lacoste-Julien. Gradient-based neural DAG learning. In 8th International Conference on Learning Representations, ICLR 2020

- [2] Lazar Atanackovic, Alexander Tong, Jason Hartford, Leo J. Lee, Bo Wang, and Yoshua Bengio. DynGFN: Bayesian Dynamic Causal Discovery using Generative Flow Networks, February 2023.

- [3] Patrik Reizinger, Yash Sharma, Matthias Bethge, Bernhard Scholkopf, Ferenc Huszar, and Wieland Brendel. Jacobian-based Causal Discovery with Nonlinear ICA. Transactions on Machine Learning Research, April 2023.

- [4] Shohei Shimizu, Patrik O Hoyer, Aapo Hyvärinen, Antti Kerminen, and Michael Jordan. A Linear NonGaussian Acyclic Model for Causal Discovery. Journal of Machine Learning Research, 7(10), 2006.

**Limitations:**

The authors provide an **honest comparison** of their method in the experiments, clearly stating its limitations compared to other methods.

---

> ### Author Rebuttal · Authors · 2024-08-06
>
> We thank the reviewer for the thorough review, the constructive comments and the stimulating question. We will go through these points in what follows.
>
> ## Weaknesses
>
> We thank the reviewer for the constructive comments: we propose to implement all the 4 main points suggested, as we agree they will be valuable improvements in the flow of the discussion. We add two specific remarks:
>
> - we already provide a definition of *active path* in Definition 5 (L 485-486 “a path $\pi$ […] is **active** w.r.t. …”). For reference, the definition is taken from “Causal Reasoning with Ancestral Graphs”, 2008, Zhang; though we agree that using “a path that is not blocked” in the main text simplifies the life of the reader, as this is more commonly used.
> - For the residuals estimation (eq. 12) we use kernel ridge regression, as proposed in NoGAM original paper. Note the target function in least squares regression is the expectation of the target given the covariates, i.e. $\mathbb{E}[Y|X]$ for generic X,Y variables, where Y is the target. This is consistent with our theoretical analysis, where the residuals of the score $\partial_{X_j} \log p(X)$ in equation 12 are defined as $\partial_{X_j} \log p(X) - \mathbb{E}[\partial_{X_j} \log p(X) | R_j]$ which are indeed the least squares regression residuals. We agree that the discussion about residual estimation is relevant, and we will add it in the main text of the paper.
>
> Concerning the minor points raised, we also agree that they would make a valuable addition, we implement all points up to the last. For a title that better conveys the meaning of the paper we will consider something in the flavor of “Identifiability of latent variable additive noise models with score matching”
>
> We are eager to mention the reviewer’s contribution in improving the clarity of our work in the acknowledgments of our paper.
>
> ## Questions
>
> This question is very interesting. We will start focusing on [1], and then will consider [3, 4], as these are the three papers we are familiar with (and where I hope I can provide a compelling answer).
>
> 1. Comment about connections to [1] (GraN-DAG): one intuition about score methods for causal discovery is that they are based on the idea that if we could access the individual kernels composing the Markov factorization, we would have all the information to recover the causal relations (assuming identifiability as in restricted ANMs). Intuitively, this is achieved as follows
>     - Take the logarithm to transform the Markov factorization in a summation
>     - Take the partial derivatives of this log-likelihood: partial derivatives at one time (i) isolate the kernel of leaf nodes (i.e. find the causal direction, as in our Propositions 2 and 3) (ii) inform about the dependency of each kernel from the other variables (i.e. can replace conditional independence tests, our Proposition 1).
>
>     GraN-DAG uses the Jacobian to do the equivalent of (ii), i.e. replace conditional independence testing with sparsity of the Jacobian to find some kind of connection strength (see equation 15 of their paper). The reason why no logarithm appears is that while in our case we go through score matching estimation to try to access kernels of the markov factorization, they model such kernels directly with a neural network.
>
>
>    This connection between their Equation 15 and Spantini et al., 2017 is not explicit in their work, neither in any existing paper, to the best of our knowledge, so **we believe that this analysis would make a valuable addition to our paper**, as a discussion in the appendix. Moreover, one interesting conjecture we can make based on these considerations is that there may be a possibility to generalize GraN-DAG to do neural causal discovery in confounded scenarios with the same guarantees that we provide. Please take this with a grain of salt.
>
> 2. Connection with [3,4]: as [3] is the nonlinear version of ICA-LiNGAM proposed in [4], we believe that connections about one work would also apply to the other. In general, the two works deal with different types of Jacobian, compared to ours or to the Jacobian found in [1]: [3, 4] deal with Jacobian of the inverse transform (linear and nonlinear), whereas our work concerns the Jacobian of the score. Despite this note, we find some point of contact may exist: Lemma A.1 of [3] proposes a condition to detect the presence of a directed path from $X_i$ to $X_j$ based on the condition $\frac{\partial f_j(X)}{\partial N_i} \neq 0.$ This is not immediately related to our Jacobian of the score, where we can identify the Markov network by checking $\frac{\partial}{\partial X_i X_j} \log p(X)$: though, a closer relation appears in the nonlinear Gaussian additive noise model case, where for a sink node we have $\frac{\partial}{\partial X_i X_j} \log p(X) = -\frac{1}{\sigma_j^2}\frac{\partial f_j(X)}{\partial X_i}$ and edges are detected by the condition $-\frac{1}{\sigma_j^2}\frac{\partial f_j(X)}{\partial X_i} \neq 0$ (note the similarity with the condition of Lemma A.1 in [3]). As they consider the partial derivative with respect to noise $N_i$, they can find direct paths, while as we consider partial derivatives with respect to $X_i$ , we can find direct edges (this statement has subtleties: the one we notice here is that we find a direct edge if and only if we know that $X_j$ is a sink, else we simply find an edge in the Markov network). Please refer to Lemma 1 in [Montagna et al., 2023a] for the derivation of the latter expression and all relevant details. We can conjecture that, in  SCMs more general than nonlinear Gaussian ANM, both our method and their methods find a way to use partial derivatives of the mechanisms to probe the presence of edges in the graph. This may also be something worth adding and expanding in our paper, for a profound view on the connection between our work and existing methodologies.

---

> > ### Comment · Reviewer_WsKx · 2024-08-07
> > **Score increased 7->8**
> >
> > Thank you for your detailed and thoughtful response. My concerns are addressed, I increase my score 7->8.

---

### Official Review · Reviewer_TL8k · 2024-07-12

**Soundness:** 3
**Presentation:** 3
**Contribution:** 1
**Rating:** 3
**Confidence:** 4

**Summary:**

The authors propose AdaScore, a method for causal discovery that generalizes previous work based on score matching for SCMs with possibly latent nodes. They combine connections of the score to conditional independence as well as to additive noise SCMs and show that a NoGAM-type procedure works to recover the direction of non-confounded edges of the corresponding partial ancestral graph.

**Strengths:**

- Adapting NoGAM [1] to the case allowing hidden variables is a practically meaningful contribution.
- Model assumptions are somewhat weakened compared to CAM-UV by allowing for general mechanisms within blocks of observed and latent parents.

**Weaknesses:**

The novelty of the paper lies primarily in the application of NoGAM to orient very specific edges in a partial ancestral graph (PAG) (the ancestral graph that represents the Markov equivalence class, in analogue to a CPDAG). This falls significantly short of the main contributions as described by the authors on l33--55. Specifically:

- The authors state that they show how constraints on the Jacobian of the score can be used as conditional independence testing. However, the extent to which this is done is only by noticing the equivalence between conditional independence and the corresponding zero in the Jacobian term (previously noted in [1,2]), without any formal analysis of the proposed t-test (Appendix C) as a statistical test of conditional independence (which happens to be a notoriously difficult test).

- The authors state that their identification results for additive noise models generalize the previous results obtained by previous works. In l193, the authors state "we remove the nonlinearity assumption (of [3]) and make the weaker hypothesis of a restricted additive noise model", but 1) this is a stronger assumption than additive noise, not a weaker one, and 2) the authors in [3] also consider the same restricted additive noise model.

- The authors claim that AdaScore is able to handle a broad class of causal models (l54), but three out of four possible situations are direct applications of existing work. 1) Under no structural assumptions with or without latent confounders, AdaScore simply performs constraint-based causal discovery (FCI) using the conditional independence properties of the Jacobian of the score, a straightforward application of [1] also previously noticed in [2]. 2) Under an additive noise assumption, AdaScore is exactly equivalent to NoGAM. 3) Only under an additive noise assumption with hidden confounders, does AdaScore generalize NoGAM to orient unconfounded edges of the PAG returned by FCI, which may be very few of the discovered adjacencies.

### Other comments

- The experiments do not seem to suggest that AdaScore out performs other methods in any meaningful way---in fact, in Figure 1 a) AdaScore is completely equivalent to NoGAM, and is thus redundant. In Figure 1 b), where AdaScore should distinguish itself, it does not appear to be consistently better than CAM-UV.

- Much of the paper (> 5 pages) is spent on directly describing previous works, NoGAM[3] and/or provide basic background on DAGs and MAGs.

[1] Spantini et al., "Inference via low-dimensional couplings." JMLR 2018.
[2] Montagna et al., "Scalable causal discovery with score matching." CLeaR 2023.
[3] Montagna et al., "Causal discovery with score matching on additive models with arbitrary noise." CLeaR 2023.

**Questions:**

- Is it correct to say that the unconfounded edges that are oriented by AdaScore are the purely _undirected_ edges of the PAG, and not the potentially bidirected ones?

**Limitations:**

The authors do not adequately discuss the limitations of their method---the limitations section in the appendix focuses purely on the empirical study. The authors claim that AdaScore is adaptive in the sense of being "less reliant on prior assumptions which are often untestable", but this is only in the sense that it performs different algorithms depending on user specification, which hardly constitutes one single unifying adaptive algorithm.

---

> ### Author Rebuttal · Authors · 2024-08-06
>
> We thank Reviewer TL8k for their thorough review and the valuable comments therein. One important criticism exposed by the reviewer is that the contribution of our work is limited in relation to the existing literature: in this regard, we address to general response and the first point of our response in the section below.
>
> ## Weaknesses
>
> - *“The authors claim that AdaScore is able to handle a broad class of causal models (l54), but three out of four possible situations are direct applications of existing work. …“.* In reply to points 1) and 2) that are made, the fact that 3 out of 4 possible situations **are extension of existing works**, does not mean that the 4th one is not there. Our main contribution and the relation with previous work, are largely discussed in the paper and in the rebuttal, particularly in the **first point of the general response**.
>
>    Point 3) of the review says that the only novel part “may only apply to few edges.” We politely but firmly disagree with this statement: the number of edges whose direction is theoretically identifiable by AdaScore but not identified by methods such as FCI depends on the problem at hand: there is no ground to overlook this because of the subjective belief that this does not happen frequently. The reviewer agrees that there is a novel contribution w.r.t. the existing literature (from the review: “*Adascore does generalize NoGAM to orient unconfounded edges of the PAG returned by FCI”;*). The reviewer also acknowledge that this is not done by any other existing methods (throughout the review, it is mentioned that our method, and thus our theory, is more general than CAM-UV, FCI, and NoGAM. We mention that it is more general than RCD and LiNGAM, also). To these points we add that our work is the first one that provides identifiability theory for latent variables causal discovery with score matching, a recent and lively line of work (Montagna et al. (a, b, c), Amin et al., Zhu et al., Rolland et al.)  discussed in the paper (L35) with connection to the field of causal representation learning Varici et al. (2024). The only point raised which may undermine our contribution is that our theory and method are not relevant given that the identifiability of the causal direction “*may apply to few edges”,* which is a purely subjective consideration, as it depends on the problem at hand and can not be simply disregarded. **Given these remarks, we kindly ask the reviewer to reconsider their score on the contribution, and the global score, accordingly.**
>
> - *“The authors state that they show how constraints on the Jacobian of the score can be used as conditional independence testing […]”.* We do not state that the Jacobian of the score can be used as conditional independence testing or that we propose a statistical test. We say that we use the Jacobian of the score **as an alternative to conditional independence testing** (L9-10, L41-42): while constraint-based methods (e.g., PC) usually rely on conditional independence testing (e.g. Zhang et al., 2011), instead, we consider sparsity of the Jacobian of the score. These are two different statistical problems. Our task concerns the estimation of the Jacobian of the score (Rolland et al. for details on such estimation, Zhu et al. for its statistical efficiency, both references are in the paper), and using a t-test for finding entries of such Jacobian with zero means: the t-test is not part of our novel contributions, we will add a citation e.g. to Walpole et al., (1972) for relative details.
>
>     Later, the reviewer writes “[…] *the extent to which this is done is only by **noticing** the equivalence between conditional independence and the corresponding zero in the Jacobian term (noted in [1, 2])*”: **we don’t merely note** the equivalence. **We prove** **it** as a corollary of Spantini et al., (see the 1st point of the general response). Concerning the relation with [2],referenced  in our paper, they proposed a version of our Proposition 1 in the limited setting of nonlinear additive Gaussian noise models, while our result holds under generic SCM.
>
>
> - *“In L193, the authors state, "we remove the nonlinearity assumption (of [3]) and make the weaker hypothesis of a restricted additive noise model", but 1) this is a stronger assumption [...]”*  [3] assume a *nonlinear restricted additive noise model,* while we assume a *restricted additive noise model.* Allowing for linear and nonlinear mechanisms is weaker than allowing for nonlinear mechanisms only.
>
> ## Limitations
>
> We remark the adaptivity of AdaScore in the sense described in the paper, meaning that it can perform with theoretical guarantees on latent variable models, still identifying the causal direction when this is theoretically possible, which is not done by other methods (FCI, NoGAM, LiNGAM), or it is done under more restrictive assumptions (CAM-UV, RCD). We remark that this does not require any user specification. The user specification simply allows to turn off the search of latent variables, if this is thought as unnecessary. This is all described in the main paper (see L263-265, and L266 where “We […] describe the version of our algorithm whose output is a mixed graph”, i.e. AdaScore when the user is willing to account for latent variables). We could easily remove the possibility of interacting with AdaScore to provide prior knowledge, but this would be a poorer design choice (e.g. see [DoDiscover](https://www.pywhy.org/dodiscover/dev/tutorials/markovian/example-pc-algo.html#Define-the-context) for a discussion about this point from an existing library).
>
>
> We __renew our thanks to the reviewer__ for the points raised, and are willing to incorporate suggestions to better reflect the nature of our contribution, naming Proposition 1 “Corollary 1”, which better reflects its link with Spantini et al. If our response is satisfactory relative to the points raised, we kindly ask the reviewer to reconsider their score to our paper.

---

> > ### Comment · Reviewer_TL8k · 2024-08-12
> >
> > I would like to thank the authors for their response. However, the rebuttal does not sufficiently address my main concerns with the paper, and I will opt to retain my score at this time.
> >
> > ## On the main contribution of the paper.
> >
> > > the fact that 3 out of 4 possible situations are extension of existing works, does not mean that the 4th one is not there.
> >
> > As mentioned in the original review, I acknowledge the novelty of using score matching to orient edges in a mixed graph. I think I am in agreement with the authors that this is the main contribution of the work. In view of this, the paper does not do a thorough enough investigation to warrant publication at this time. The specific example I mentioned pertains to the lack of an analysis of the proposed t-test as a conditional independence test.
> >
> > >  We do not state that the Jacobian of the score can be used as conditional independence testing or that we propose a statistical test. We say that we use the Jacobian of the score as an alternative to conditional independence testing (L9-10, L41-42): while constraint-based methods (e.g., PC) usually rely on conditional independence testing (e.g. Zhang et al., 2011), instead, we consider sparsity of the Jacobian of the score.
> >
> > If using the Jacobian of the score as an alternative to a conditional independence test relieves it from being analyzed as a statistical test, then additional theory should be developed, e.g., adapting Thm. 3.2 from Zhu et al. on the theoretical properties of the discovered graph. This may not be a trivial addition, as Zhu et al. deals only with the non-linear Gaussian model.
> >
> > > we don’t merely note the equivalence. We prove it as a corollary of Spantini et al., (see the 1st point of the general response).
> >
> > As reviewer `mmww` also points out, this is a rather trivial application of the result when assuming faithfulness.
> >
> > ## On adaptiveness of the algorithm
> >
> > > We remark the adaptivity of AdaScore in the sense described in the paper, meaning that it can perform with theoretical guarantees on latent variable models, still identifying the causal direction when this is theoretically possible, which is not done by other methods (FCI, NoGAM, LiNGAM), or it is done under more restrictive assumptions (CAM-UV, RCD).
> >
> > This is closer to my understanding of the contribution of the paper, but  not how I understood the authors interpretation of "adaptivity". For example, in l261: "The main strength of our approach is its adaptivity with respect to structural assumptions: based on the user’s belief about the plausibility of several modeling assumptions on the data".
> >
> > > We remark that this does not require any user specification. The user specification simply allows to turn off the search of latent variables, if this is thought as unnecessary.
> >
> > I do not understand how this is not user specification. For me, an algorithm that is adaptive in the sense of the paper would need a routine to detect the presence of (significant) latent confounders, then automatically apply the mixed graph version. Otherwise, the AdaScore algorithm reads more as a computer application that includes previous methods rather than a unifying algorithm.
> >
> > ### More minor points.
> >
> > > Allowing for linear and nonlinear mechanisms is weaker than allowing for nonlinear mechanisms only.
> >
> > My apologies, thank you for the clarification.
> >
> > > Point 3) of the review says that the only novel part “may only apply to few edges.”
> >
> > I acknowledge that the "few edges" comment was subjective. I recognize that this may have come off as dismissive and I apologize for that. This was however not a significant point for my score.

---

> > > ### Author Response · Authors · 2024-08-13
> > >
> > > We are surprised by the request for finite samples guarantees on the score matching estimation, as this request was not in the original review. Overall it is clear that we disagree with the reviewer about this paper. In any case, learning theory results are far beyond the scope of the paper: the majority of causal discovery algorithms in the literature do not come with finite sample guarantees.
> > >
> > > Concerning the use of the word adaptivity: we agree with the reviewer that we could adopt a better wording; we will adopt the phrasing used in the rebuttal to better express the benefits of our method.

---

### Official Review · Reviewer_mmww · 2024-07-12

**Soundness:** 3
**Presentation:** 3
**Contribution:** 2
**Rating:** 4
**Confidence:** 4

**Summary:**

The paper extends theoretical results about causal discovery through score matching to encompass both linear and non-linear SCMs and lift the sufficiency assumption. The theoretical results relax the non-linearity assumption of Montagna et al 2023 by swapping it with the less restrictive one of restricted ANM from Peters et al 2009. As for the latent confounder detection a parallel with m-separation is drawn using results from Spantini et al 2018, to establish that the score will be non-zero in the presence of an active path. Following the theoretical results, an algorithm to estimate causal graphs from data is proposed and evaluated, generalizing the NoGAM algorithm of Montagna et al 2023, which only covers the non-linear case.

**Strengths:**

The paper is clearly written and, if it wasn’t for some of the definitions relegated to the appendix, very easy to follow.

The theoretical results, particularly Propositions 2 and 3 are important extensions of the score-matching methodology for causal discovery, dealing with both the non-linearity and sufficiency assumptions of the method proposed in Montagna et al. 2023.

**Weaknesses:**

The paper motivation is basically the weakening of current assumptions for causal discovery methods. However, assumptions and benefits of the proposed methodology are not clearly specified. In line 74, the authors state that faithfulness is assumed (I believe, it is kind of hidden in the background notions). If that is the case, the method adopts the same assumption as FCI, plus ANM. So the proposed method is relaxing assumptions compared to CAM-UV, RCD and NoGAM, but adding onto FCI. Regarding the benefits, the alleged flexibility of the method to output DAGs, MECs, MAGs, PAGs, which should make it preferable to FCI, is merely touched upon in the contributions and the experiment section.

Proposition 1 is a rather trivial application of the more general lemma in Spantini et al. and it does not specify the required faithfulness assumption to obtain the result from Eq. 6 in the paper.

The experimental results show limited added value according to the one metric chosen (SHD) in a synthetic setting. They are not comprehensive enough, with no application to common (pseudo-)real benchmarks (e.g. from bnlearn). More experiments and more metrics are needed as, as it stands, the proposed method seems to add no real value compared to the baselines. Additionally, it is not clear from the experiments if it is really able to identify confounders. Breaking down precision and recall by mark would show this. FCI and a random baseline should be also added for reference.

Experiments are conducted on data with at most 9 variables, and the scalability of the method is not shown nor discussed.

The model used to estimate residuals is not discussed, nor the assumption that the chosen model fits the data adequately to correctly estimate residuals, and what is needed to assess this.

**Questions:**

- Line 28: FCI weakness is highlighted to be that it outputs PAGs, does the proposed method improve on this at parity of assumptions?

- Line 44: ANM is an assumption on the causal mechanism, is it not?

- Line 217: "the noise terms recentered by the latent causal effects" what does this mean?

- Line 221: "N_i are assumed to be non-Gaussian when f_i is linear" This reduces to RCD, right?

- Line 257: "the remaining arrowheads … are identified no better than in the equivalence class" what does this mean?

- Line 279: "using hypothesis testing to find vanishing MSE..." how is this set up?

- Line 280: how much added value does pruning have? E.g. CAM pruning can have huge impact https://arxiv.org/pdf/1310.1533

- Alg. 1: do you start from a disconnected graph? Where do the "remaining" bidirected edges come from, do you reduce them prior to this?

- Line 292: exogenous variables are selected at random, hence could be sink nodes with no consequence on the discovery process. Alternative and more meaningful testing strategies could be to select confounders (e.g. https://proceedings.neurips.cc/paper_files/paper/2021/file/144a3f71a03ab7c4f46f9656608efdb2-Paper.pdf)

- Line 305: "presents better performance" I am not sure about this, they seem pretty aligned considering quartiles. Statistical tests to ascertain this are missing.

- Line 315: some realistic scenarios are available through e.g. the bnlearn repository (https://www.bnlearn.com/bnrepository/). Any reason not to test on some of the data available there?

Minor points and typos

- Line 73: reference for d-separation is missing

- Line 79: reference for MEC is missing

- Line 130: have -> has

- Line 147: "directed global Markov property": not introduced. I guess it is just the global Markov condition introduced before?

- Footnote 1: provides -> provide

- Line 162: "unobserved active paths" are not defined in Definition 5, m-separation.

- Line 209: full stop missing

- Line 236: the PAG -> a PAG

- Line 267: "Appendix C.2" what should the reader expect to find in there?

- Line 296: "We fix the […] level" significance

**Limitations:**

yes

---

> ### Author Rebuttal · Authors · 2024-08-06
>
> We thank the reviewer for the time and effort in reading our paper. One important concern unaddressed by our general response is about the limits of our experimental evaluation: we point to the first bullet in our response below, and the experimental results in the PDF of the rebuttal.
>
> ## Weaknesses
>
> - “*The experimental results show limited added value according to the one metric chosen (SHD) in a synthetic setting. They are not comprehensive enough …*” We highly regard this suggestions and will include real data experiments in our paper. Our results on bnlearn datasets presented in the PDF of the rebuttal (Fig. 2) show that AdaScore always outperforms CAM-UV and RCD, outperforms NoGAM on 3 out of 4 datasets, and outperforms LiNGAM on 2 out of 4 datasets. Thus AdaScore is a practical alternative with better theoretical identifiability guarantees than any of the other methods we consider. Though, please note that real and synthetic benchmarks for causal structure learning are documented as far from being satisfactory, see e.g. “Self-Compatibility: Evaluating Causal Discovery without Ground Truth”, 2024, Faller et al; experimental results should be taken with a grain of salt.
> - *“In line 74, the authors state that faithfulness is assumed (I believe, it is kind of hidden in the background notions).*” In L74 we write “*we assume that the reverse direction* $X_i \perp X_j | X_Z \implies X_i \perp^d_{\mathcal{G}} X_j | X_Z$ *hold*”: this is the definition of the faithfulness assumption (see our references therein), so the faithfulness assumption is not hidden in background notion, but explicitly made for the first time in L74.
>
>    ”*If that is the case, the method adopts the same assumption as FCI, plus ANM. …*”. We point to the second bullet of the general response for the answer on this point. Concerning the request of an experimental comparison with FCI, consider that the two methods have different identifiability guarantees and so different types of graphs as output. In this sense, FCI is not the ground for a fair comparison, as it loosens the assumptions at the price of weaker identifiability guarantees. We propose to compare the FCI F1 accuracy in skeleton recovery versus AdaScore F1 accuracy in skeleton recovery when inference is done according to Proposition 1 (at the base of the skeleton inference in AdaScore), which does not require assumptions on the SCM. Experimental results are shown in the PDF of the rebuttal (Fig. 3): we see that AdaScore consistently outperforms FCI in the task of recovery of the PAG skeleton.
>
> - “*Proposition 1 is a rather trivial application of the more general lemma in Spantini et al. and it does not specify the required faithfulness assumption to obtain the result from Eq. 6*” For the first part (that Proposition 1 is a trivial application of Spantini et al.) we point to the first bullet in the general response of this rebuttal. For the second point made about Spantini et al. and the faithfulness assumptions, some clarifications are needed: Eq. 6 in the paper is the Lemma of Spantini et al. (and not obtained from it, as written in the review) and does not require faithfulness assumption. To avoid any confusion, **we propose to write** “*Note that this result does not make use of faithfulness assumption”* right after Eq. 6. Proposition 1 is a corollary of Spantini et al. that makes use of the faithfulness assumption.
> - *“[…] scalability of the method is not shown nor discussed.”* Scalability is discussed in Appendix C.3 and Proposition 5, also referenced in the main manuscript, L 284. We propose to add experiments with larger number of variables to the camera-ready version.
> - *“The model used to estimate residuals is not discussed, nor the assumption that the chosen model fits the data adequately to correctly estimate residuals, and what is needed to assess this.”* We use kernel ridge regression, as proposed in NoGAM original paper. Note the the target function in least squares regression is the expectation of the target given the covariates, i.e. $\mathbb{E}[Y|X]$ for generic X,Y variables, where Y is the target. This is consistent with our theoretical analysis, where the residuals of a variable $V_k$ are defined as $V_k - \mathbb{E}[V_k|V_{Z\setminus \set{k}}]$ (Equation 15) which are indeed the least squares regression residuals. We agree that this point is worth discussing, and will add it to the main text of our paper.
>
> ## Questions
>
> - *“Line 28 …”* As causal directions can not be identified at parity of assumptions, we do not improve upon FCI identifiability guarantees in that setting. Better identifiability comes from additional assumptions, as discussed in the general response.
> - *Line 44 …* Yes, thank you.
> - *”Line 217: ...*" “Recentered” is intended in the sense that we have the random variable $f_i(V_{PA_i})$, the observed causal effects, that are shifted by $g_i(U^i)$ addition (see equation 13). We will provide a clearer formulation of this point.
> - *”Line 221: ..."* Although this certainly subsumes the setting of RCD, it does not necessarily. We can have a mixture of linear and non-linear dependencies between $X_i$ and its observable parents and also arbitrary relations to the unobserved parents.
> - *”Line 257: ...”* Our method can provide no further orientations than the ones that already follow from the Markov-equivalence class. We will write this clearer form in the paper.
> - *”Line 279: ...”* See section C.2
> - *”Line 280 ...* We also observed that pruning has a non-negligible impact. We will add the ablation study in the camera-ready version of the paper.
> - *”Alg. 1: do you start from a disconnected graph? ...* As we describe more explicitly in algorithm 2, we start by adding undirected edges between $X_i$ and $X_j$ if $X_i \perp X_j | X_V \setminus \{X_i, X_j\}$. We will make this more clear in the main text in the updated version.
> - *“Line 305: ...* We will add statistical tests to the empirical evaluation in the camera ready version.

---

> > ### Comment · Reviewer_mmww · 2024-08-08
> >
> > Many thanks for the clarifications and the extra experiments provided. Responding in order:
> >
> > - Thanks for providing the results on some of the bnlearn datasets. I agree on having to take them with a pinch of salt. Indeed I would create CI around the numbers you provided, by changing the seed of the BN DGP. By the looks of the plot, I believe that a lot of them could not be significantly different from each other.
> >
> > - I apologise if my comment about the assumption being "kind of hidden" came across the wrong way. I was in no way accusing. I was just pointing out that, given the importance of assumptions in your claim about the added value of your proposed method, the organisation and wording of the paper do not make clear enough their importance, strictness or benefits. In my view, the way you explained in the second bullet of the general rebuttal is already much clearer and effective, at a high level, than the paper. As for the comparison with FCI, the skeleton accuracy is good. The way I would go about it is to transform the output mixed graphs from ADAscore to a PAG, then compare to FCI. Otherwise you only compare less than half of the work done by the algorithm.
> >
> > - The point to stress is not that Eq 6 does not use faithfulness, it's that Proposition 1 does.
> >
> > - Re scalability, thanks to the pointer to the appendix, I had indeed seen that. I still think that the discussion is not satisfactory. The main paper just reads: "This way, we get an algorithm that is polynomial in the best case (Appendix C.3)." and, to me, that is not a great way to discuss potential limitations. Why only 9 variables? I would show a comparison of elapsed/computing time to back up your claim that the proposed algorithm is practical and with superior guarantees.
> >
> > - Line 28: great, now it is clear, thanks. Again, from a reader perspective, the way the motivation is presented raises expectations that the paper does not fulfill.
> >
> > - Line 44: in my view, the paper reads like ANM is not an assumption worth noting. To go back to my previous point about "kind of hidden", assumptions should be better organised and elaborated upon.
> >
> > - Line 279: Thanks for the pointer. I had not read that since you only reference it when talking about the pruning. I would separate the two discussions, title them more clearly and reference them accordingly in the main paper.
> >
> > - Line 280: interesting, the reference I provided show a large impact of pruning.
> >
> > - Line 305 and extra experiments provided: I still think that the experiments show that the method is not really good at doing what it is supposed to do better than the others algorithms: for linear systems with confounders it is basically the same as random and lingam is much better (though, of course, it is more specialised). For the rest of the scenarios it is almost always at par or worse than other baselines. Confounder detection is quite poor, as shown in Figure 4 in the additional results.

---

> > > ### Author Response · Authors · 2024-08-09
> > >
> > > We thank the reviewer for dedicating time to our rebuttal.
> > >
> > > ---
> > >
> > > **Q1** “*Indeed I would create CI around the numbers you provided.”* We agree with the reviewer relative to the need to produce confidence intervals: we commit to adding them in the camera-ready.
> > >
> > > ---
> > >
> > > **Q2**
> > > - *”the way you explained in the second bullet of the general rebuttal is already much clearer”* We will adopt this wording and add explicit assumption paragraphs.
> > >
> > > - *“The way I would go about it is to transform the output mixed graphs from ADAscore to a PAG, then compare to FCI.”* We find the required experiment not as straightforward as it seems. The output of AdaScore a priori only makes statements about direct causal connections but not about d-separations (or m-separations). E.g. suppose our algorithm outputs $X \leftrightarrow Y \leftrightarrow Z$. This means we cannot orient the edge between X and Y and Y and Z, which could mean that either there is a hidden mediator between X and Y (and the edge could go in either way, i.e. $X \rightarrow L \rightarrow Y$ or $X \leftarrow L \leftarrow Y$) or a hidden confounder. In the former case, we could have both, $X \perp Z | Y$ and $X \not\perp Z | Y$. Therefore the PAG w.r.t. our output is not well defined. Thus, to test “the other half” of our work a more sensible approach is via experiments as in Figure 1 of the paper.
> > >
> > > ---
> > >
> > > **Q3.** “*Re scalability …”*
> > >
> > > - The pointer to Appendix C.3 is indeed to better discuss the scalability, as the content of the appendix would not fit in the main paper page limit.
> > > - On scalability experiments
> > >     1. Fig 5 of the paper already shows AdaScore scalability with the number of samples.
> > >     2. We propose to add more experiments on AdaScore scalability with the number of nodes: we present preliminary results showing that AdaScore scales much better than CAM-UV, the method with best theoretical guarantees previous to ours. The values in the table are mean +- std in seconds.
> > >
> > >     |  | 3 nodes | 5 nodes | 7 nodes | 9 nodes |
> > >     | --- | --- | --- | --- | --- |
> > >     | adascore | 1.555 ± 0.274 | 12.582 ± 6.568 | 43.596 ± 29.867 | 133.950 ± 81.151 |
> > >     | camuv | 3.999 ± 2.340 | 17.703 ± 3.894 | 80.196 ± 31.040 | 198.977 ± 69.306 |
> > >     | nogam | 2.374 ± 0.129 | 6.127 ± 0.631 | 12.042 ± 0.608 | 20.407 ± 0.796 |
> > >     | rcd | 0.409 ± 0.330 | 2.507 ± 0.960 | 10.720 ± 3.774 | 22.798 ± 3.641 |
> > >     | lingam | 0.015 ± 0.0 | 0.051 ± 0.001 | 0.124 ± 0.001 | 0.251 ± 0.001 |
> > >
> > > ---
> > > **Q4 (Line 28)** *“great, now it is clear, thanks. Again, from a reader's perspective, the way the motivation is presented raises expectations that the paper does not fulfill.”* Reviewer originally asked whether we could identify more than FCI, at parity of assumptions: as we discussed, this is theoretically impossible (see Glymour et al.) Thus, stating “*The FCI algorithm [11] can only return an equivalence class from the data. Appealing to additional restrictions ensures the identifiability of some direct causal effects in the presence of latent variables*” as done in L28-30 does not raise unfulfilled expectations, as this is exactly what we do in our paper.
> > >
> > > ---
> > > **Q5 (Line 44)** “*in my view, the paper reads like ANM is not an assumption worth noting.”*
> > >  L44 we write “*we prove that the score function identifies the causal direction of ANMs, with minimal assumptions on the causal mechanisms”,* the *minimal* part refers to additivity of the noise: this is a minimal requirement to achieve identifiability in the light that most of existing works place linearity/nonlinearity assumptions on the mechanisms.
> > >
> > > We will remove the word minimal, stating “*we prove that the score function identifies the causal direction under the assumption of additive noise models, without further requirements on the causal mechanisms.”* We thank the reviewer to help us clarifying this point.
> > >
> > > ---
> > >
> > > **Q6 (Line 305)**
> > > - *“for linear systems with confounders it is basically the same as random and lingam is much better”* Out of the 16 experimental settings of Fig. 1 of the PDF rebuttal, AdaScore is comparable to random only in one or two - large-scale linear. LiNGAM is comparable to random in at least 5 out of 16 (see nonlinear).
> > > - *“it is almost always at par or worse than other baselines”* Considering median performance (please, recall that we already committed to adding statistical tests in the main paper) we see CAM-UV better than AdaScore 4 out of 16 times, NoGAM 2 out of 16, RCD 4 out 16, LiNGAM 4 out of 16.
> > >
> > > Reviewer's claim that other methods are better than AdaScore is not supported by empirical evidence, and it does not account that their performance is achieved via stricter assumptions. Additionally, we remark on the following points that emerged in the discussion agreed upon both by us and the reviewer
> > >
> > > 1. Conclusions from experiments must be drawn with care
> > > 2. Our baseline has better theoretical guarantees, which are very relevant as they are valid beyond the scope of our experiments, i.e. in the real world.

---

> > > > ### Comment · Reviewer_mmww · 2024-08-09
> > > >
> > > > Q2
> > > >
> > > > I agree that my proposed strategy is less straightforward than I anticipated given the different meanings of the mixed graph output of your method. I frankly cannot think of a solution right now but I take your point that you did what you could in the timeframe.
> > > >
> > > >
> > > > Q3
> > > >
> > > > I am well aware of the space limitation, I was not asking the authors to move C.3. I was pointing out that the current content of the paper regarding scalability is not satisfactory in my view. I still believe that. In fact, the table you provide about elapsed time (thanks) already shows that the actual performance has a very exponential trend. I was just asking for clarity.
> > > >
> > > > Fig 5 in the paper shows shd. Number of samples is different from numeber of nodes.
> > > >
> > > >
> > > >
> > > >
> > > >
> > > > Q6 (Line 305)
> > > >
> > > > I am afraid but the burden of proof falls on you here, not me. I am not claiming anything, I am just supposing, and trying to interpret some unclear results. Again, what I am trying to elicit here is clarity, given that the authors do not provide readers with the means (e.g. statistical tests) to validate their claims.
> > > >
> > > > The author's claim that "CAM-UV better than AdaScore 4 out of 16 times, NoGAM 2 out of 16, RCD 4 out 16, LiNGAM 4 out of 16" is unsubstantiated until they present tests to back up it up. The overlap in distributions shown in Figure 1 make me suppose that the statistical differences between pairs of methods will be very minimal. But again, it is a supposition since I do not have the means to judge.

---

> > > > > ### Author Response · Authors · 2024-08-10
> > > > >
> > > > > We thank the reviewer for promptly engaging in the discussion.
> > > > >
> > > > > - **Q3.** Fig 5: we apologise for the confusion. We will add a similar plot with the elapsed time on the y-axis.
> > > > > - **Q6.** We did not intend that the reviewer needs to prove results about the goodness of Adascore, and we apologize for this confusion: our quantitative statements were aimed at rebutting that AdaScore “*is almost always at par or worse than other baselines”,* as the empirical evidence does not support this. To help the analysis of our empirical results, we are willing to provide statistical tests in the time frame of the authors reviewers discussion. In particular, we would consider Whitney Mann-U test, testing for each algorithm and node size
> > > > >     1. the null-hypothesis that the distributions are identical against the alternative that adascore has a stochastically lower distribution (i.e. lower median, e.g. assuming the average SHD of adascore and the competitor algorithm and adascore have the same distribution up to shift in location, in our case ~ Gaussian by CLT)
> > > > >     2. The null-hypothesis that the distributions are identical against the alternative that adascore has a stochastically greater distribution (i.e. greater median)
> > > > >
> > > > >     If the reviewer finds this a satisfactory approach,  we will do our best to provide these results in time to discuss them with the reviewer. Otherwise, we kindly ask the reviewer to specify what statistical tests they had in mind for the comparison.

---

> > > > > > ### Comment · Reviewer_mmww · 2024-08-10
> > > > > >
> > > > > > Thanks for the availability and commitment to provide clear and robust results for the evaluation of your work.
> > > > > >
> > > > > > If you are willing to make the extra assumption (of equal distributions up to a shift in location) in order to interpret the rank WM test as a difference in medians, the same assumption would make a one-sided t-test on the difference in means also valid. In fact, WM is usually used to test differences for asymmetric distributions, which, if you assume gaussianity, do not apply. A t-test would also consider dispersion around the mean, rather than the rank only.
> > > > > >
> > > > > > In any case, I would find a WM test satisfactory if you already have it setup. I am not trying to make the authors' life difficult, and again, I appreciate the efforts.

---

> > > > > > > ### Author Response · Authors · 2024-08-12
> > > > > > > **Statistical tests**
> > > > > > >
> > > > > > > We ran WM statistical tests on the SHD relative to the 32 experimental settings presented in Figure 1 (dense graphs) and Figure 3 (sparse graphs) of the main paper. We summarize the statistical tests outcomes in the following table: in the “*Alternative: less”* column we provide the number of times we reject the null in favor of the alternative that AdaScore SHD is stochastically lower (i.e. AdaScore "better"), in the “*Alternative: greater”* column we provide the number of times we reject the null in favor of the alternative that AdaScore SHD is stochastically greater (i.e. AdaScore "worse"). As level of the test to reject the null, we consider 0.1.
> > > > > > >
> > > > > > > &nbsp;
> > > > > > >
> > > > > > > |  | Alternative: Less | Alternative: Greater |
> > > > > > > | --- | --- | --- |
> > > > > > > | LiNGAM | 14/32 | 6/32 |
> > > > > > > | RCD | 14/32 | 2/32 |
> > > > > > > | NoGAM | 2/32 | 0/32 |
> > > > > > > | CAM-UV | 3/32 | 1/32 |
> > > > > > > | Random | 29/32 | 0/32 |
> > > > > > > | Fully Random | 29/32 | 0/32 |
> > > > > > > &nbsp;
> > > > > > >
> > > > > > > These __results suggest the following interpretations__:
> > > > > > >
> > > > > > > - None of the alternative methods is preferable to AdaScore
> > > > > > > - In absence of prior knowledge about the SCM underlying data generation, AdaScore is preferable in terms of SHD accuracy to LiNGAM, RCD, and the random baselines
> > > > > > > - In absence of prior knowledge about the SCM underlying data generation, we can not say much in comparison to NoGAM and CAM-UV.
> > > > > > >
> > > > > > > These outcomes match our findings in the paper, where we summarise that AdaScore “accuracy is often comparable and sometimes better than competitors.” (L 311-312); findings that are now more substantiated, for which we credit the reviewer suggesting a better experimental analysis. Note that these results are achieved while having the least strict assumptions among all of the benchmarked methods. __We propose to add these results to the main paper__, to substantiate our claims on empirical performance. Additionally, we remark on our intention to add all the additional experiments already discussed with the reviewer and partially presented in the PDF of the rebuttal.
> > > > > > >
> > > > > > > For the sake of completeness, we present the table of p-values of all the MW tests in separate comments (due to space limits). As these contain valuable information, we propose to include barplots with binned range to visualize the different p-values associated to different algorithms in the main paper, such that results interpretation is not dependent on the choise on the level of the test.

---

> > > > > > > > ### Author Response · Authors · 2024-08-12
> > > > > > > > **Tables of p-values for sparse graphs**
> > > > > > > >
> > > > > > > > For completeness, we present the table of p-values of all the MW tests on experiments relative to __sparse graphs__. Please notice that adding the random and fully random baselines to the experiments changed the random state of our running relative to Fig. 3 in the main paper. Results are qualitatively the same, but we observe a few mismatches due to the variance resulting from different random states.
> > > > > > > >
> > > > > > > > # Alternative: less
> > > > > > > >
> > > > > > > > __linear Fully observable model__
> > > > > > > > |              |           3 |           5 |           7 |           9 |
> > > > > > > > |:-------------|------------:|------------:|------------:|------------:|
> > > > > > > > | camuv        | 0.5147    | 0.6020    | 0.2644    | 0.8236    |
> > > > > > > > | nogam        | 0.5147    | 0.4852    | 0.1575      | 0.05256   |
> > > > > > > > | rcd          | 0.3696    | 0.7820    | 0.0216   | 0.6578    |
> > > > > > > > | lingam       | 0.5147    | 0.8762    | 0.8425      | 0.9999    |
> > > > > > > > | random       | 5.41254e-06 | 5.41254e-06 | 5.41254e-06 | 5.41254e-06 |
> > > > > > > > | fully_random | 5.41254e-06 | 5.41254e-06 | 5.41254e-06 | 5.41254e-06 |
> > > > > > > >
> > > > > > > >
> > > > > > > > __nonlinear Fully observable model__
> > > > > > > > |              |           3 |           5 |           7 |           9 |
> > > > > > > > |:-------------|------------:|------------:|------------:|------------:|
> > > > > > > > | camuv        | 0.2893    | 0.2893    | 0.4267    | 0.09515   |
> > > > > > > > | nogam        | 0.4852    | 0.2179    | 0.1575      | 0.3152    |
> > > > > > > > | rcd          | 5.41254e-06 | 5.41254e-06 | 1.08251e-05 | 0.0003  |
> > > > > > > > | lingam       | 0.0034  | 0.0034  | 0.0177   | 0.0446   |
> > > > > > > > | random       | 5.41254e-06 | 5.41254e-06 | 5.41254e-06 | 5.41254e-06 |
> > > > > > > > | fully_random | 5.41254e-06 | 5.41254e-06 | 5.41254e-06 | 5.41254e-06 |
> > > > > > > >
> > > > > > > >
> > > > > > > > __linear Latent variables model__
> > > > > > > > |              |           3 |           5 |           7 |           9 |
> > > > > > > > |:-------------|------------:|------------:|------------:|------------:|
> > > > > > > > | camuv        | 0.5147    | 0.6847    | 0.4852   | 0.8912    |
> > > > > > > > | nogam        | 0.4852    | 0.6303    | 0.5732    | 0.2644    |
> > > > > > > > | rcd          | 0.3696    | 0.3979    | 0.1237    | 0.3979    |
> > > > > > > > | lingam       | 0.6578    | 0.8425      | 0.7593    | 0.9942    |
> > > > > > > > | random       | 2.16502e-05 | 2.16502e-05 | 0.000162376 | 5.41254e-06 |
> > > > > > > > | fully_random | 1.08251e-05 | 5.41254e-06 | 5.41254e-06 | 5.41254e-06 |
> > > > > > > >
> > > > > > > >
> > > > > > > > __nonlinear Latent variables model__
> > > > > > > > |              |          3 |           5 |           7 |           9 |
> > > > > > > > |:-------------|-----------:|------------:|------------:|------------:|
> > > > > > > > | camuv        | 0.3421   | 0.1763    | 0.5441    | 0.2406    |
> > > > > > > > | nogam        | 0.1965   | 0.5147    | 0.217936    | 0.0987    |
> > > > > > > > | rcd          | 0.0044 | 0.0025  | 0.0525  | 0.0177   |
> > > > > > > > | lingam       | 0.0715  | 0.0116   | 0.1399    | 0.0057  |
> > > > > > > > | random       | 0.0003 | 1.08251e-05 | 6.49505e-05 | 0.0001 |
> > > > > > > > | fully_random | 0.0003 | 5.41254e-06 | 2.16502e-05 | 2.16502e-05 |
> > > > > > > >
> > > > > > > > ---
> > > > > > > > &nbsp;
> > > > > > > >
> > > > > > > > # Alternative: greater
> > > > > > > >
> > > > > > > > __linear Fully observable model__
> > > > > > > > |              |        3 |        5 |        7 |           9 |
> > > > > > > > |:-------------|---------:|---------:|---------:|------------:|
> > > > > > > > | camuv        | 0.5147 | 0.4267 | 0.7593 | 0.1965    |
> > > > > > > > | nogam        | 0.5147 | 0.5441 | 0.8762 | 0.9553    |
> > > > > > > > | rcd          | 0.6578 | 0.2644 | 0.9855 | 0.3979    |
> > > > > > > > | lingam       | 0.5147 | 0.1399 | 0.1763 | 0.0001 |
> > > > > > > > | random       | 1        | 1        | 1        | 1           |
> > > > > > > > | fully_random | 1        | 1        | 1        | 1           |
> > > > > > > >
> > > > > > > >
> > > > > > > > __nonlinear Fully observable model__
> > > > > > > > |              |        3 |        5 |        7 |        9 |
> > > > > > > > |:-------------|---------:|---------:|---------:|---------:|
> > > > > > > > | camuv        | 0.7355 | 0.7355 | 0.6020 | 0.9172 |
> > > > > > > > | nogam        | 0.5441 | 0.8236 | 0.8600 | 0.7106 |
> > > > > > > > | rcd          | 1        | 1        | 0.9999 | 0.9997 |
> > > > > > > > | lingam       | 0.9974 | 0.9974 | 0.9855 | 0.9684 |
> > > > > > > > | random       | 1        | 1        | 1        | 1        |
> > > > > > > > | fully_random | 1        | 1        | 1        | 1        |
> > > > > > > >
> > > > > > > >
> > > > > > > > __linear Latent variables model__
> > > > > > > > |              |        3 |        5 |        7 |          9 |
> > > > > > > > |:-------------|---------:|---------:|---------:|-----------:|
> > > > > > > > | camuv        | 0.5147 | 0.3696 | 0.5441 | 0.1399   |
> > > > > > > > | nogam        | 0.5441 | 0.4267 | 0.4558 | 0.7593   |
> > > > > > > > | rcd          | 0.6578 | 0.6578 | 0.9048 | 0.6303   |
> > > > > > > > | lingam       | 0.3696 | 0.1965 | 0.2893 | 0.0092 |
> > > > > > > > | random       | 0.9999 | 0.9999 | 0.9999 | 1          |
> > > > > > > > | fully_random | 0.9999 | 1        | 1        | 1          |
> > > > > > > >
> > > > > > > >
> > > > > > > > __nonlinear Latent variables model__
> > > > > > > > |              |        3 |        5 |        7 |        9 |
> > > > > > > > |:-------------|---------:|---------:|---------:|---------:|
> > > > > > > > | camuv        | 0.6847 | 0.8425   | 0.5147 | 0.8034 |
> > > > > > > > | nogam        | 0.8236 | 0.5147 | 0.8236 | 0.9048 |
> > > > > > > > | rcd          | 0.9965 | 0.998  | 0.9623 | 0.9855 |
> > > > > > > > | lingam       | 0.9384 | 0.9907 | 0.8912 | 0.9965 |
> > > > > > > > | random       | 0.9997 | 0.9999 | 0.9999 | 0.9999 |
> > > > > > > > | fully_random | 0.9997 | 1        | 0.9999 | 0.9999 |

---

> > > > > > > > > ### Comment · Reviewer_mmww · 2024-08-13
> > > > > > > > >
> > > > > > > > > Thanks for providing the test results. They go in the direction that I feared.
> > > > > > > > > This is what I read from the tables provided (I focus on the dense graphs and 9 variables as, in my view, 10 variables graphs are a minimum of usefulness\practicality and usually a minimum in the literature too, $\alpha=0.05$):
> > > > > > > > > - linear fully observable: on par with random and the others, worse than lingam (expected)
> > > > > > > > > - linear latent: worse than camuv, rcd and lingam
> > > > > > > > > - nonlinear full: better than rcd and lingam (expected)
> > > > > > > > > - nonlinear latent: better than lingam (expected)
> > > > > > > > > - all the rest are not significantly different
> > > > > > > > >
> > > > > > > > > Overall, the results could be summarised as follows: Adascore performs better than lingam in the nonlinear case but worse in the linear one. **In the linear latent case, the focus of the proposed extension, it performs worse than all baselines apart the very akin nogam.** Hence, although the stronger guarantees, empirical performance shows that Adascore is mostly on par with prior methods, only better than a method specialised in linear SCM, and worse than others in the scenario where it provides the added guarantees.
> > > > > > > > >
> > > > > > > > > Overall, the discussion highlighted the following points:
> > > > > > > > > - Assumptions need to be clarified
> > > > > > > > > - Contributions need to be clarified
> > > > > > > > > - Scalability issues need to be clarified
> > > > > > > > > - Experimental results do not show much added value
> > > > > > > > >
> > > > > > > > > I believe the above constitutes enough rework to need additional reviews. I therefore intend to keep my score. I thank the authors for the good discussion.

---

> ### Author Response · Authors · 2024-08-12
> **Tables of p-values for dense graphs**
>
> For completeness, we present the table of p-values of all the MW tests on experiments relative to __dense graphs__. Please notice that adding the random and fully random baselines to the experiments changed the random state of our running relative to Fig. 1 in the main paper. Results are qualitatively the same, but we observe a few mismatches due to the variance resulting from different random states.
>
> # Alternative: less
>
> __Linear fully observable model__
> |              |           3 |           5 |           7 |         9 |
> |:-------------|------------:|------------:|------------:|----------:|
> | camuv        | 0.657895    | 0.544101    | 0.485256    | 0.782064  |
> | nogam        | 0.657895    | 0.342105    | 0.139931    | 0.455899  |
> | rcd          | 0.514744    | 0.782064    | 0.514744    | 0.8425    |
> | lingam       | 0.657895    | 0.803476    | 0.999756    | 1         |
> | random       | 0.000243564 | 6.49505e-05 | 0.000525017 | 0.315264  |
> | fully_random | 1.08251e-05 | 1.08251e-05 | 5.41254e-06 | 0.0216285 |
>
> __nonlinear Fully observable model__
> |              |           3 |           5 |           7 |           9 |
> |:-------------|------------:|------------:|------------:|------------:|
> | camuv        | 0.139931    | 0.240625    | 0.0177315   | 0.098781    |
> | nogam        | 0.397968    | 0.369682    | 0.602032    | 0.630318    |
> | rcd          | 3.78878e-05 | 0.000243564 | 2.16502e-05 | 5.41254e-06 |
> | lingam       | 0.0177315   | 0.00927169  | 0.0315064   | 0.00446535  |
> | random       | 0.000102838 | 1.08251e-05 | 1.08251e-05 | 1.08251e-05 |
> | fully_random | 6.49505e-05 | 5.41254e-06 | 5.41254e-06 | 5.41254e-06 |
>
> __linear Latent variables model__
> |              |         3 |           5 |        7 |         9 |
> |:-------------|----------:|------------:|---------:|----------:|
> | camuv        | 0.485256  | 0.823659    | 0.544101 | 0.982269  |
> | nogam        | 0.264424  | 0.782064    | 0.289371 | 0.1575    |
> | rcd          | 0.426714  | 0.630318    | 0.735576 | 0.985597  |
> | lingam       | 0.455899  | 0.139931    | 0.982269 | 0.998955  |
> | random       | 0.0715701 | 0.000525017 | 0.139931 | 0.0715701 |
> | fully_random | 0.037628  | 0.000525017 | 0.176341 | 0.426714  |
>
> __nonlinear Latent variables model__
> |              |        3 |          5 |          7 |          9 |
> |:-------------|---------:|-----------:|-----------:|-----------:|
> | camuv        | 0.735576 | 0.426714   | 0.684736   | 0.426714   |
> | nogam        | 0.630318 | 0.217936   | 0.630318   | 0.455899   |
> | rcd          | 0.973787 | 0.0525612  | 0.759375   | 0.426714   |
> | lingam       | 0.289371 | 0.00574812 | 0.0216285  | 0.00927169 |
> | random       | 0.123725 | 0.0019431  | 0.0116153  | 0.00342073 |
> | fully_random | 0.176341 | 0.00036264 | 0.00927169 | 0.00342073 |
>
> ---
> &nbsp;
>
> # Alternative: greater
>
> __linear Fully observable model__
> |              |        3 |        5 |           7 |           9 |
> |:-------------|---------:|---------:|------------:|------------:|
> | camuv        | 0.369682 | 0.514744 | 0.544101    | 0.240625    |
> | nogam        | 0.369682 | 0.684736 | 0.876275    | 0.602032    |
> | rcd          | 0.514744 | 0.264424 | 0.544101    | 0.196524    |
> | lingam       | 0.369682 | 0.240625 | 0.000525017 | 5.41254e-06 |
> | random       | 0.999897 | 0.999978 | 0.999637    | 0.710629    |
> | fully_random | 1        | 0.999995 | 1           | 0.982269    |
>
>
> __nonlinear Fully observable model__
> |              |        3 |        5 |        7 |        9 |
> |:-------------|---------:|---------:|---------:|---------:|
> | camuv        | 0.891219 | 0.782064 | 0.988385 | 0.917253 |
> | nogam        | 0.630318 | 0.657895 | 0.455899 | 0.426714 |
> | rcd          | 0.999989 | 0.999838 | 0.999995 | 1        |
> | lingam       | 0.988385 | 0.992655 | 0.973787 | 0.997402 |
> | random       | 0.999935 | 0.999995 | 1        | 0.999995 |
> | fully_random | 0.999962 | 1        | 1        | 1        |
>
>
> __linear Latent variables model__
> |              |        3 |        5 |         7 |         9 |
> |:-------------|---------:|---------:|----------:|----------:|
> | camuv        | 0.544101 | 0.196524 | 0.514744  | 0.026213  |
> | nogam        | 0.759375 | 0.264424 | 0.759375  | 0.876275  |
> | rcd          | 0.602032 | 0.426714 | 0.315264  | 0.0177315 |
> | lingam       | 0.573286 | 0.891219 | 0.0216285 | 0.0019431 |
> | random       | 0.938497 | 0.999637 | 0.891219  | 0.938497  |
> | fully_random | 0.968494 | 0.999756 | 0.860069  | 0.602032  |
>
>
> __nonlinear Latent variables model__
> |              |        3 |        5 |        7 |        9 |
> |:-------------|---------:|---------:|---------:|---------:|
> | camuv        | 0.315264 | 0.602032 | 0.342105 | 0.630318 |
> | nogam        | 0.426714 | 0.823659 | 0.426714 | 0.602032 |
> | rcd          | 0.037628 | 0.955395 | 0.289371 | 0.630318 |
> | lingam       | 0.759375 | 0.996579 | 0.985597 | 0.992655 |
> | random       | 0.904842 | 0.99856  | 0.992655 | 0.998057 |
> | fully_random | 0.860069 | 0.999756 | 0.994252 | 0.997402 |

---

> ### Author Response · Authors · 2024-08-13
>
> We thank the reviewer for the active participation in the discussion. We are surprised that the reviewer specifies only now that all experiments with less than ten variables and in sparse settings are not meaningful in analyzing the algorithm performance, as this point was never raised in the review or in the authors-reviewers discussion, where the requests were limited to a better analysis of scalability (without even mentioning density as a discriminating point). Regarding the claim that these are the setting common in the literature, we point to CAM-UV paper (the work closest to ours), where they have experiments on at most nine variable, and one semi synthetic experiment on ten variables. Further, we notice that the analysis of the reviewer on the goodness of AdaScore compared to other methods is not based on statistically significant statements: e.g. CAM-UV is better than AdaScore with statistical significance in one experimental setting out of 32. Yet, the reviewer concludes that CAM-UV is better than AdaScore on inference on linear latent variables models, which encompasses 8 out of 32 of our experimental settings. As the statistical tests were an explicit and well-motivated request from the reviewer to be able to make statistically significant conclusions, we will stick to the conclusions supported by the statistical tests.
>
> In this spirit, we provide the summary table with the level of the test $0.05$, as asked by the reviewer:
>
> |  | Alternative: Less | Alternative: Greater |
> | --- | --- | --- |
> | LiNGAM | 13/32 | 6/32 |
> | RCD | 12/32 | 2/32 |
> | NoGAM | 0/32 | 0/32 |
> | CAM-UV | 1/32 | 1/32 |
> | Random | 26/32 | 0/32 |
> | Fully Random | 29/32 | 0/32 |
>
> These results contrast with the analysis of the superiority of certain methods made by the reviewer. Specifically, we respond to specific points made by the reviewer and not supported by the empirical evidence:
>
> - “Linear fully observable on par with random”: we have the following list of p-values for *alternative : less* on linear fully observable models comparing with random: 0.0002, 6.49e-05, 0.0005, 0.31, 5.41e-6, 5.41e-6, 5.41e-6, 5.41e-6. The only case in which p-value is higher than 0.05 is 0.31. The *alterantive : greater* pvalues instead are never below 0.05 threshold for random, and not even close: 1, 1, 1, 1, 0.99, 0.99, 0.99, 0.71. AdaScore is not on par with random.
> - “linear latent: worse than camuv”: as already discussed, there is only one time in which *alternative :great* in comparison with CAM-UV, in the linear latent setting, goes below 0.05. The claim is not supported by empirical evidence
>
> Further, the reviewer writes that the linear latent case is the focus of our work: we disagree, both the linear and nonlinear settings are equally important, as the point of the paper is indeed relaxing linearity and nonlinearity assumptions on the causal mechanisms, in the context of latent variable models. If we suggested that linear latent variable cases were the focus of our work, we kindly ask the reviewer to point to the specific parts of the paper.
>
> Finally, concerning the claim that contributions need to be clarified, we again point to L48, the paragraph starting with “our main contributions” written in bold. Concerning the claim that assumptions need to be clarified, the only specific criticism expressed on this point was relative to the positioning of faithfulness assumption, that could be more upfront. We already addressed this and remark on our offer to make a separate paragraph.

---

### Author Rebuttal · Authors · 2024-08-06

We thank the reviewers for the time spent reading and understanding our paper, as well as for the insightful comments and questions. Our paper is **well received in terms of clarity** - with Presentation scores 3 from all reviewers - **and soundness** - with scores 3, 3, 4 from R TL8k, R mmww, R  WsKx respectively. A more **polarized view concerns the amount of contribution** of our work, with score ranging from 1 to 4. R mmww and R WsKx find Proposition 3 an important contribution, whereas  R TL8k recognizes the novelty in our result concerning the identifiability of directed edges in restricted ANM with latent variables but argues that in practice “very few of the discovered adjacencies” happen to be identifiable by our rule. We will go into the details of this in the individual response. In the PDF attached to the rebuttal, we present experimental results with a random baseline (Fig. 1), experimental results on bnlearn data (Fig. 2), a comparison with FCI performance (Fig. 3), and experiments with the F1 metric (Fig. 4), asked by R mmww.

In the general response, we address the __two concerns that are common to R TL8k and R WsKx__, which are the lack of novelty in Proposition 1 and the relation of our method relative to FCI.

- Proposition 1 is a corollary of an existing result from Spantini et al.: in our work, we extensively discuss the related result of Spantini et al., explicitly mentioning it as the main resource to derive Proposition 1 (which we specify to be *Adapted from Spantini et al.,* see Proposition 1 statement). To make this point clearer and tune down our contributions in the results of Proposition 1, we propose to call it Corollary 1, making its relation with Spantini et al. even more explicit. In addition to this, in our work we explicitly state that this is not our main contribution, and write that our main contribution (Proposition 3) builds on the existing results of Spantini et al., and Montagna et al., L46-47 “*On these results, we build the main contribution of our work*”. Specifically, in the subsequent paragraph (L48) we specify that our main contribution is that (i) we provide theoretical ground for score based causal discovery of hidden variables and identifiable direct causal edges in latent variable models and (ii) we translate these findings in an algorithm. In this sense, **we kindly ask the reviewer to consider what we explicitly state to be the main contributions** in the paper, which do not include Proposition 1.
- Concerning the relation to FCI, reviewers highlight how it requires fewer assumptions, compared to our method: it is indeed the case that we make the same assumption of FCI plus ANM. The point that is elaborated in the paper is that the ANM assumption (in addition to those made, e.g., by FCI) allows us to prove the identifiability of the direction of certain causal edges in the PAG, that are not identifiable under the FCI assumption. The trade off between identifiability and assumptions is well known, as it is intrinsic to the dichotomy between constraint-based and functional-based causal discovery,  see e.g. “Review of Causal Discovery Methods Based on Graphical Models”, 2019, Glymour et al. Overall, our solution provides identifiability guarantees that are not given by any method in the literature that is based on restrictions of the SCM, which are the main ground of comparison of our work, as they belong to the same category of functional-based methods. Lastly, we answer the question from R TL8k asking whether “*Is it correct to say that the unconfounded edges that are oriented by AdaScore are the purely undirected edges of the PAG, and not the potentially bidirected ones?*”: in this context, to further tell apart our method from FCI we highlight the following, stronger, identifiability guarantees of AdaScore compared to FCI:
    - AdaScore can direct unconfounded edges that are undirected in FCI.
    - AdaScore can direct unconfounded edges that are potentially bidirected in FCI (i.e. $\circ \hspace{-1.4mm}\rightarrow$). An easy example is the three variable collider $X \rightarrow Y \leftarrow Z$. FCI will output the graph $X \circ \hspace{-1.4mm}\rightarrow Y \leftarrow \hspace{-1.4mm} \circ Z$, i.e. potentially bidirected edges. If there are no hidden confounders, under restricted ANM assumptions our method is able to orient both edges. Since the semantics of our directed edges imply that there are no hidden confounders, we also know that these edges would have tails at $X$ and $Z$ in the PAG semantics. This answers R TL8k question directly.
    - Directed edges in AdaScore encode direct causal effects, whereas directed edges in FCI only encode ancestral relationships.

    Note that at parity of assumptions, the empirical results presented in the PDF of the rebuttal show that AdaScore outperforms FCI in the  inference of the PAG skeleton (Fig. 3 of the PDF)


**References of the rebuttal**

Scalable and flexible causal discovery with an efficient test for adjacency, 2024, Amin et al.

Scalable Causal Discovery with Score Matching, 2023a, Montagna et al.

Causal Discovery with Score Matching on Additive Models with Arbitrary Noise, 2023b, Montagna et al.

Assumption violations in causal discovery and the robustness of score matching, 2023c, Montagna et al.

Score matching enables causal discovery on additive noise models, 2022, Rolland et al.

Inference via low-dimensional couplings, 2017, Spantini et al.

Probability and statistics for engineering and scientists, 1972, Walpole et al.

Score-based Causal Representation Learning: Linear and General Transformations, Varici et al., 2024

Kernel-based Conditional Independence Test and Application in Causal Discovery, 2011, Zhang et al.

---

### Decision · Program_Chairs · 2024-09-25

**Decision:**

Reject

**Comment:**

This manuscript was actively discussed at length. First, with the authors during the rebuttal period, and afterwards, again among the reviewers and myself. It is fair to say that all reviewers agree that the manuscript puts forward an interesting idea, and everyone also appreciated that the authors did their due diligence during the rebuttal by providing additional results. Considering the manuscript, reviews, and discussion, there remain a number of concerns that withhold me from recommending acceptance at this time. First and foremost, the authors claim the adaptivity and flexibility of their method is unmatched, although the experiments do not support that the proposed method convincingly outperforms the baselines esp. in scenarios where it should. Second, originally raised by reviewer mmww, scalability of the method to larger numbers of variables is a major concern; the authors show equivalance of sparsity of the Jacobian to that of conditional independence, and as we all know, methods based on the latter often suffer in terms of scalability. Third, the manuscript at times leans towards overselling. It would certainly benefit from a clear discussion of the actual contributions (e.g. in light of existing work such as NoGAM) and in particular the limitations of the method (including the two previous points). All in all, I recommend the authors to improve their manuscript in these regards, and I am looking forward to seeing the improved version published at a later date.